# CORRELATED DENSE ASSOCIATIVE MEMORIES

## ABSTRACT

We introduce a novel associative memory model named *Correlated Dense Associative Memory* (CDAM), which integrates both auto- and hetero-association in a unified framework for continuous-valued memory patterns. Employing an arbitrary graph structure to semantically link memory patterns, CDAM is theoretically and numerically analyzed, revealing four distinct dynamical modes: auto-association, narrow hetero-association, wide hetero-association, and neutral quiescence. Drawing inspiration from inhibitory modulation studies, we employ anti-Hebbian learning rules to control the range of hetero-association, extract multi-scale representations of community structures in graphs, and stabilize the recall of temporal sequences. Experimental demonstrations showcase CDAM's efficacy in handling real video data and replicating a classical neuroscience experiment.

## 1 INTRODUCTION

### 1.1 BACKGROUND

Mathematical models of ferromagnetism in statistical mechanics, as developed by Lenz, Ising, Schottky, and others (Brush, 1967; Folk & Holovatch, 2022), model the interactions between collections of discrete variables. When connected discrete variables disagree in their values, the energy of the system increases. The system trends toward low energy states via recurrent dynamics, but can be perturbed or biased by external input. Marr (1971) proposed a conceptual framework of associative memory in neurobiological systems using a similar principle but of interacting neurons, which was subsequently formalised in a similar way (Nakano, 1972; Amari, 1972; Little, 1974; Stanley, 1976; Hopfield, 1982)[1]. A key difference between these associative memory and ferromagnetism models is that the neurons are typically connected all-to-all with infinite-range interactions whereas in the ferromagnetism models variables were typically connected locally within a finite range.

The principle by which these associative memory networks store memories is by assigning recurrent connection weights and update rules such that the energy landscape of the network forms dynamic attractors (low energy states) around memory patterns (particular states of the neurons). In the case of pairwise connections, these weights translate to the synaptic strength between pairs of neurons in biological neural networks. The network therefore acts as a content addressable memory – given a partial or noise-corrupted memory, the network can update its states through recurrent dynamics to retrieve the full memory.

Of particular interest to the machine learning community is the recent development of dense associative memory networks (Krotov & Hopfield, 2016) (also referred to as modern Hopfield networks) and their close correspondence (Ramsauer et al., 2021) to the attention mechanism of Transformers (Vaswani et al., 2017). In particular, the dense associative memory networks introduced by Krotov & Hopfield (2016) (including with continuous variables) were generalised by using the softmax activation function, whereby Ramsauer et al. (2021) showed a connection to the attention mechanism of Transformers (Vaswani et al., 2017). Indeed, Krotov & Hopfield (2016) make a mathematical analogy between their energy-based update rule and setwise connections given their energy-based update rule can be interpreted as allowing individual pairs of pre- and post-synaptic neurons to make multiple synapses with each other – making pairwise connections mathematically as strong as equivalently-ordered setwise connections. Demircigil et al. (2017) later proved this analogy to be

---

[1]Simultaneously, work in spin glasses followed a similar mathematical trajectory in the works of Sherrington & Kirkpatrick (1975) and Pastur & Figotin (1977)

accurate in terms of theoretical memory capacity. As shown subsequently, by explicitly modelling higher-ordered connections in such networks, the energy landscape becomes sharper and memory capacity is increased (Burns & Fukai, 2023).

In the majority of the prior associative memory works discussed so far, memory recall is auto-associative, i.e., given some partial memory the dynamics of the network ideally lead to recalling the (same) full memory. However, hetero-association is just as valid dynamically (Amari, 1972; Gutfreund & Mezard, 1988; Griniasty et al., 1993; Gillett et al., 2020; Tyulmankov et al., 2021; Millidge et al., 2022; Karuvally et al., 2023; Chaudhry et al., 2023)[2]: instead of a partial memory directing the dynamics to recalling the same memory pattern, we can instead recall something else. Such hetero-associations are believed to naturally occur in the oscillatory dynamics of central pattern generators for locomotion (Stent et al., 1978), sequence memory storage in hippocampus (Treves & Amit, 1988), and visual working-memory in primate temporal cortex (Miyashita, 1988).

## 1.2 MOTIVATIONS FROM NEUROSCIENCE

A classical result in the hetero-association neuroscience literature is due to Miyashita (1988). This work demonstrated hetero-association of stimuli in monkey temporal cortex could arise semantically via repeated presentations of the same stimuli in the same order, not only spatially via similarities in the stimuli themselves. Miyashita (1988) showed neurons responsive to presentation of randomly-generated fractal patterns had a monotonically-decreasing auto-correlation between the firing rates due to the current pattern and the next expected patterns, up to a distance of 6 patterns into the future.

Work on numerosity in birds, non-human primates, and humans (Nieder et al., 2002; Ditz & Nieder, 2015; Nieder, 2012; Kutter et al., 2018) have repeatedly provided evidence of neurons responding to specific numbers or quantities. In these experiments, the stimuli (numbers or quantities) can be both semantically and spatially correlated – i.e., they can have the known semantic ordering of $1, 2, 3 \ldots$ or 'some, more, even more $\ldots$', as well as the spatial or statistical relationships between the stimuli. Notably, even in abstract number experiments where spatial correlations are moot, semantic distances up to a range of $\sim 5$ numbers[3] (as measured by significant auto-correlations of the neural activity) are common.

This phenomenon extends beyond simple 1D, sequence relationships, however. Schapiro et al. (2013) presented human participants with a series of arbitrary visual stimuli which were ordered by a random walk on a graph with community structure (where each image was associated with a vertex in the graph). Functional magnetic resonance imaging analysis of the blood-oxygen-level-dependent response showed the representations of different stimuli were clustered by brain activity into the communities given by the underlying graph and unrelated to the actual stimuli features.

In all of these studies, both auto-association (for the present stimulus) and hetero-association (for the semantically-related stimuli) is present. And such mixtures, where they encode a more general structures relevant for tasks, may be behaviourally useful. For instance, mice trained on goal-sequence tasks sharing a common semantic basis arising from a 2D lattice graph develop task-progress cells which generalise across tasks, physical distances, behavioural timescales, and stimuli modality (El-Gaby et al., 2023). Furthermore, such dynamics may be modulated by inhibitory signals (King et al., 2013; Honey et al., 2017; Hertäg & Sprekeler, 2019; Haga & Fukai, 2019; 2021; Burns et al., 2022; Tobin et al., 2023) to shift the locus of attention, learning, or behaviour. Such function could account for the many instances of anti-Hebbian learning found throughout neural systems (Roberts & Leen, 2010; Shulz & Feldman, 2013), as well as their implications in the role of sleep for memory pruning (Crick & Mitchison, 1983; Hopfield et al., 1983; Diekelmann & Born, 2010; Poe, 2017; Zhou et al., 2020), motor control learning (Nashef et al., 2022), dendritic selectivity (Hayama et al., 2013; Paille et al., 2013) and input source separation (Brito & Gerstner, 2016).

---

[2]An interesting alternative or supplementary technique is to use synaptic delays to generate such sequences (Tank & Hopfield, 1987; Kleinfeld & Sompolinsky, 1988; Karuvally et al., 2023), however here we will focus on non-delayed hetero-association where synapses all operate at the same timescale.

[3]Depending on the species, brain area, and stimuli modality.

### 1.3 MOTIVATIONS FROM MACHINE LEARNING

Given the storied history of classical hetero-associative modelling work, extensions to dense associative memory are a natural next step. Some work in this direction has already begun. Millidge et al. (2022) present an elegant perspective which makes it straightforward to construct dense associative memory networks with hetero-association, and demonstrated recalling the opposite halves of MNIST or CIFAR10 images. Karuvally et al. (2023) construct an adiabatically-varying energy surface to entrain sequences in a series of meta-stable states, using temporal delays for memories to interact via a hidden layer. Application to a toy sequence episodic memory task showed how the delay signal can shift the attractive regime. And, recently, Chaudhry et al. (2023) studied a sequence-based extension of the dense associative memory model by adopting the polynomial or exponential update rule for binary-valued sequences of memories. This work also introduces a generalisation of the Kanter & Sompolinsky (1987) pseudoinverse rule to improve distinguishability between correlated memories. As Chaudhry et al. (2023) concludes, many potential research avenues remain, including extending these methods to continuous-valued patterns.

Chaudhry et al. (2023) also note the potential to study different network topologies. There are several distinct notions of network topology which we could study, including that of neuronal connections (as in Löwe & Vermet (2011); Burns & Fukai (2023)), spatial or statistical relationships between memory patterns (as in Löwe (1998); De Marzo & Iannelli (2023)), or semantic relationships between memory patterns (as in Amari (1972); Chaudhry et al. (2023)). A majority of classical work has focused on semantic correlation, likely due to its relevance to neuroscience (see Subsection 1.2). To extend the study of such semantic relationships to interesting topologies, it is necessary to introduce a basic topology, such as via embedding memories in graphs (as in Schapiro et al. (2013)). Being highly versatile mathematical structures, upon generalising semantic relationships with graphs, this additionally generates opportunities to study graph-based computations such as community detection or graph segmentation and learning or simulation of (finite) automata by neural networks (Balle & Maillard, 2017; Ardakani et al., 2020; Liu et al., 2023).

Section 3.5 of Millidge et al. (2022) describes how we may generally consider the relationships between auto- and hetero-associative models, and notes how Transformers' attention mechanisms take the hetero-associative form mathematically. Functionally, however, the attention mechanism is not obligated to perform hetero-association, since its values and keys are created independently by their respective weight matrices (see Vaswani et al. (2017)) and can in-principle make these identical so as to perform auto-association, or otherwise some mixture of auto- and hetero-association. Taking this perspective seriously opens the way for analysing Transformers through the lense of potential mixtures of auto- and hetero-associative dynamics, à la the analysis of a large language model in Ramsauer et al. (2021) by considering the implied energy landscapes in each of its attention heads. For this to be possible, however, a first step is to rigorously develop and study a dense auto- and hetero-association model and its inherent computational capabilities.

### 1.4 OUR CONTRIBUTIONS

With these joint motivations from neuroscience and machine learning in mind, we

- Introduce a dense associative memory model, called *Correlated Dense Associative Memory (CDAM)*, which combines a controllable mixture of auto- and hetero-association in a single model for dynamics on continuous-valued memory patterns, using an underlying (arbitrary) graph structure to semantically hetero-associate the memory patterns;

- Theoretically and numerically analyse CDAM's dynamics, demonstrating connections to graph theory and four distinct dynamical modes – auto-association, narrow hetero-association, wide hetero-association, and neutral quiescence;

- Taking inspiration from inhibitory modulation studies, we demonstrate how anti-Hebbian learning rules can be used to: (i) widen the range of hetero-association across memories; (ii) extract multi-scale representations of community structures in memory graph structures; and (iii) stabilise recall of temporal sequences; and

- Illustrate, via experiments, CDAM's capacity to work with real video data and replicate data from a classical neuroscience experiment.

## 2 CORRELATED DENSE ASSOCIATIVE MEMORY (CDAM)

### 2.1 MODEL

To embed memories in the network, we first create $P \in \mathbb{N}$ patterns as continuous-valued vectors of length $N \in \mathbb{N}$, the number of neurons in the network. These *memory patterns* can be random, partially-random, or themselves contain content we wish to store. In the random case, each value of a memory vector is independently sampled from the interval $[0, 1]$. In the partially-random case, we reserve half of the vector for structured memory and the rest is random in the same sense as before. We denote the value for a neuron $i$ in an individual memory pattern $\mu$ as $\xi_i^\mu$. For convenience, we organise these vectors into a memory matrix $\Xi \in \mathbb{R}^{N \times P}$ for convenience. We also define a vector $\tilde{\xi} \in \mathbb{R}^N$ whose values are $\tilde{\xi}_i = P^{-1} \sum_{\mu=1}^{P} \xi_i^\mu$ to represent the average 'memory load' of each neuron.

Next, we choose a graph $M = (V, \Delta)$ with $|V| = P$ vertices and $|\Delta| \in \mathbb{N}$ edges. This graph, which we also refer to as the *memory graph*, forms the basis for the inter-pattern hetero-associations via its adjacency matrix $A$.

We use discretised time and denote the network state at time $t$ as $S^{(t)} \in \mathbb{R}^N$. To use the language of Millidge et al. (2022), we use $\mathrm{softmax}$ as our separation function, which is defined for a vector $z$ as $\mathrm{softmax}(z_i) := \frac{\exp(z_i)}{\sum_j \exp(z_j)}$. Starting at a chosen initial state $S^{(0)}$, subsequent states are given by

$$S^{(t+1)} := S^{(t)} + \eta((\mathrm{softmax}(\beta S^{(t)} \Xi)Q - N^{-1}\tilde{\xi}^T) - S^{(t)}), \quad Q = a\Xi + h(\Xi A)^T, \quad (1)$$

where $\eta \in \mathbb{R}^+$ is the magnitude of each update, $\beta \in \mathbb{R}^+$ is the inverse temperature (which can be thought of as controlling the level of mixing between memory patterns during retrieval), and $a, h \in \mathbb{R}$ is the strength of auto- and hetero-association in the retrieval projection matrix $Q$, respectively.

### 2.2 THEORETICAL ANALYSIS

A typical analysis to perform on associative memory networks is to probe its memory storage capacity, i.e., how many memories can be stored given $N$ neurons? In CDAM, when $a, h \neq 0$, the regular notions of 'capacity' seem inapplicable. This is because 'capacity' is normally measured in the pure auto-associative case by giving a noise-corrupted or partial memory pattern, and observing whether and how closely the model's dynamics converge to the uncorrupted or complete memory pattern (e.g., see Amit et al. (1985) for the classical model and Demircigil et al. (2017) for the dense model). In the pure hetero-associative case, 'capacity' has (to our knowledge) only ever been studied in the linear sequences case (e.g., see Löwe (1998) for the classical model and Chaudhry et al. (2023) for the dense model). However, in our model we study general mixtures of both auto- and hetero-association, as well as arbitrary memory graphs (not just linear cycles). It is therefore unclear whether there exists an appropriate notion of 'capacity' for this mixture.

One can, however, study the model in a similar spirit of analysis. To this end, we demonstrate the dynamics of our model in the thermodynamic limit. First, let us set aside the choices of $\eta$ and $\beta$, which control the amplitude of each step's update. For an undirected memory graph $M$, our energy function is

$$E \propto -a \sum_{\mu=1}^{P} \exp(\beta \xi^\mu S) - h \sum_{\{\alpha, \sigma\} \in \Delta(M)} \exp(\beta(\xi^\alpha S)(\xi^\sigma S)), \quad (2)$$

where $\Delta(M)$ is the set of edges in the undirected memory graph. Assume, for a moment, that $M$ is $k$–regular, meaning each vertex has degree $k$. In this case, we could rewrite Equation 2 as

$$E \propto -a \sum_{\mu=1}^{P} \exp(\beta \xi^\mu S) - h \sum_{\alpha=1}^{P} \sum_{\sigma=1}^{P} \frac{A_{\alpha,\sigma}}{k} \exp(\beta(\xi^\alpha S)(\xi^\sigma S)) \quad (3)$$

$$= -(a+hk) \sum_{\mu=1}^{P} \exp(\beta \xi^\mu S) + h \sum_{\alpha=1}^{P} \sum_{\sigma=1}^{P} \frac{A_{\alpha,\sigma}}{k} \exp(\beta(\xi^\alpha S - \xi^\sigma S)^2). \quad (4)$$

From Equation 4, we can see that while in a Hebbian hetero-associative regime, i.e., $h > 0$, setting $a < -kh$ gives the trivial minimisation of letting all $\xi^\mu S$ terms vanish, i.e., having a state which is far from any pattern. However, when $a > -kh$, minimisation of the energy demands maximising the auto-association under penalty of the consequent hetero-association. In the absence of the hetero-association penalty, we have the model of Equation 13 in Lucibello & Mézard (2023), where scaling comes from $a$; similarly, in the absence of auto-association, we have Chaudhry et al. (2023) scaled by $h$ and with arbitrary semantic correlations according to $M$. In our case, however, where there is a mixture of auto- and hetero-associations (which act simultaneously), the hetero-associative component of Equation 4 causes a large number of pattern activations for negative values of $a$, i.e., while $0 > a > -kh$. When $a, h > 0$, hetero-association remains but across a narrower range.

In Appendix A.1, we perform numerical simulations to demonstrate these four modes: auto-association, narrow hetero-association, wide hetero-association, and neutral quiescence. These simulations also demonstrate how in $k$–regular graphs we need $h = \frac{1-a}{k}$ for the mean neural activity to converge to 0 in the limit of $T \to \infty$, i.e., to keep an unbiased excitatory–inhibitory (E–I) balance[4]. And while the above analysis assumes $M$ is $k$–regular, this E–I balance finding applies for non-regular memory graphs by setting $h = \frac{1-a}{m}$, where $m = |V|^{-1} \sum_{\mu \in V(M)} \deg(\mu)$, in the limit of $P \to \infty$ and $N \to \infty$.

For the case of a directed memory graph $\overrightarrow{M}$, the energy function is

$$E \propto -\beta^{-1} \log \left( a \sum_{\mu=1}^{P} \exp(\beta \xi^\mu S) + h \sum_{(\alpha,\sigma) \in \Delta(\overrightarrow{M})} \exp(\beta(\xi^\alpha S)(\xi^\sigma S)) \right), \quad (5)$$

where $\Delta(\overrightarrow{M})$ is the set of edges in the directed memory graph. As done for Equation 4, a similar analysis for Equation 5 is possible, but is complicated by the directed edges, e.g., consider the difference between an $\overrightarrow{M}$ where all but one vertex $\mu$ point their edges to $\mu$ and an $\overrightarrow{M}$ in which each vertex has equal in- and out-degree. Relatedly, we conjecture when $\overrightarrow{M}$ is an Erdös-Renyi graph (a random graph constructed by allowing any edge with probability $p$), the critical value of $a$ which marks the transition from neutral quiescence to wide hetero-association will be proportional to $p$ when $p > \frac{(1-\varepsilon)\ln N}{N}$, i.e., when $\overrightarrow{M}$ is asymptotically connected.

**General interaction between auto- and hetero-association.** Of natural interest is when $h \neq 0$, which provides interactions between the patterns. What is interesting about Equation 1 is the possibility of both auto- and hetero-associative terms affecting the dynamics when both $a, h \neq 0$. As implied informally above, this means the model cannot perform *pure pattern retrieval*, i.e., retrieval of a single memory pattern $\xi^\mu$ without at least partial retrieval of other patterns. To show this, it is useful to refer to the alignment between a pattern $\xi^\mu$ and a state $S^{(t)}$. For this, we use the Pearson product-moment correlation coefficient, which for pattern $\mu$ at time $t$ we denote $r(\mu^{(t)})$.

**Proposition 2.1** (Hebbian auto- and hetero-associative mixtures cannot perform pure pattern retrieval for patterns not isolated in the memory graph). *Suppose $\mu$ is not an isolated vertex in $M$. Let $a, h > 0$. Then the model cannot perform pure pattern retrieval of $\xi^\mu$.*

*Proof.* Let $\{\xi^\mu, \xi^v\} \in E$ if $M$ is undirected, and let $(\xi^\mu, \xi^v) \in E$ if $M$ is directed. Setting $S^{(t)} = \xi^\mu$ will cause the second term of $Q$ to be non-negative because $h > 0$, and therefore $r(v^{(t+1)})$ will be proportionally large. Simultaneously, $r(\mu^{(t+1)})$ will be non-vanishing, since $a > 0$. Therefore, no single pattern can be purely retrieved. $\qquad \square$

---

[4]Note this does not imply there is no activity in the network, since neurons can take negative values.

**Corollary 2.2** (Pure pattern retrieval is possible for some memory graphs when the dynamics are Hebbian auto-associative or Hebbian hetero-associative, but not both). *If:*

- $a > 0$ *and* $h = 0$*; or if*

- $a = 0$*,* $h > 0$*, the out-degree of all vertices in $M$ is 1, and we have a sufficient $\beta$ and $\eta$,*

*then the model can perform pure pattern retrieval of some memory patterns.*

*Proof.* The excitatory auto-associative result, where $a > 0$ and $h = 0$, is simply a weighted version of Theorems 1–3 from Ramsauer et al. (2021). The excitatory hetero-associative result, where $a = 0$ and $h > 0$, is indicated by Proposition 2.1, with the added restriction that there exists only one memory pattern, $\xi^v$, projecting from $\xi^\mu$ in $M$. This restriction is because if the out-degree of $\xi^\mu$ was 0, then after setting $S^{(t)} = \xi^\mu$, the projection matrix $Q$ would be filled with zeroes since $a = 0$. Therefore, values of $S$ would converge to a value of $-\tilde{\xi}$. If the out-degree of $\xi^\mu$ was $> 1$, we would have the situation of Proposition 2.1, only with multiple memory patterns having large $r$ values (with their strengths proportional to the weights of their respective in-edges from $\xi^\mu$ in $M$). Finally, we need to achieve $S^{(t+1)} = \xi^v$ (or something arbitrarily close) to have pure pattern retrieval of $\xi^v$, since $a = 0$ means we will not have the luxury of additional time-steps to achieve convergence. Fortunately, by Theorem 4 of Ramsauer et al. (2021), we can get arbitrarily close by requiring sufficiently large values of $\beta$ and $\eta$ to update the state to $\xi^v$ in a single step. □

This naturally comports with Theorems 2.1 and 2.2 of Löwe (1998), wherein the classical associative memory model with binary-valued memories is studied when $M$ is a 1D Markov chain[5]. There, Löwe (1998) showed that sequence capacity increases given large semantic correlations.

**Proposition 2.3** (Connected components in an undirected memory graph are retrieved in some Hebbian hetero-associative regimes). *Let $Y \subset M$ be a finitely-sized connected component of $M$. Set $h > 0$ and $|a| < h$. Then setting $S^{(t)} = \xi^\mu$, where $\mu \in Y$, will cause $r(v^{(t+\lambda)})$ for all $v \in Y$ to be non-vanishing, for some $\lambda \in \mathbf{N}^+$ and thereafter for all time-steps.*

*Proof.* Set $S^{(t)} = \xi^\mu$. If $a = 0$, then $r(v^{(t+1)})$ will be non-vanishing for all $v \in Y$ which are are adjacent to $\mu$ in $Y$. Similarly, the adjacent vertices of those $v \in Y$ which are adjacent to $\mu$ will have non-vanishing $r(v^{(t+2)})$, and so on. If $h > a > 0$, the same argument applies. If $h < 0$, the same argument applies but the rate at which the values of $r(v)$ grow is slower. □

## 3 NUMERICAL SIMULATIONS

Now we study a wider collection of memory patterns and graphs, starting with a simple 1D cycle and gradually increasing complexity. Along the way, there are primarily two inter-weaving stories:

1. Anti-Hebbian auto-association increases the relative contribution of Hebbian hetero-association, which provides control over the range of hetero-association, extraction of multi-scale community structures in memory graphs, and stabilisation of temporal sequence recall; and

2. The flexibility of CDAM and its underlying graphical structure enables modelling a variety of phenomena, including graph community detection and sequence memory.

Unless otherwise stated, in the following numerical analyses we use $N = 1,000$, $\beta = 0.1$, and $\eta = 0.01$. We run our simulations until convergence, at which point we measure the Pearson product-moment correlation coefficient between each memory $\mu$ and the final state $S$. To initialise the network state, we choose a memory pattern $\mu$ and set $S^{(0)} = \xi^\mu + cX$, where $X$ is a random vector with elements independently drawn from the interval $[-0.5, 0.5]$ and $c \in \mathbb{R}^+$ is the amplitude of the additive random noise. Here we use $c = 1$.

---

[5]Löwe (1998) also studied the case of spatial correlations between neurons, as may arise in naturalistic data. This work has been recently continued for dense associative memory by De Marzo & Iannelli (2023).

## 3.1 CONTROLLING THE RANGE OF RECALLED CORRELATED MEMORIES

Modulating the balance of auto- and hetero-association using $a$ and $h$ allows us to control the range of memory retrieval in $M$. To demonstrate this, we use an undirected cycle graph. A *cycle graph* $C_n$ has $n$ vertices connected by a single cycle of edges through all vertices. As described and illustrated in Appendix A.2, cycle graphs are the most commonly studied semantic hetero-associative memory structure previously studied, most likely due to it being a fitting representation of temporal sequences. In Appendix A.3, we show choices of $a$ and $h$ which achieve good fits ($R^2 = 0.996$) with the experimental data reported in Miyashita (1988).

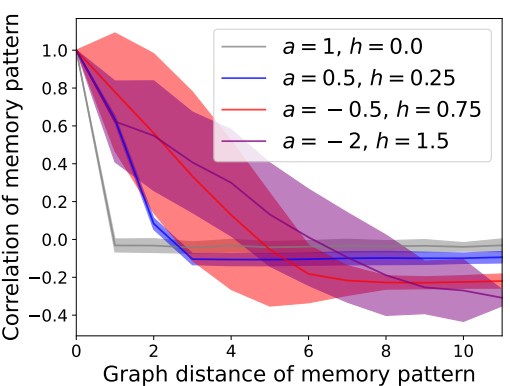

Figure 1 measures the range of the spread across values of $a$ and $h$, with significant differences observed between the tested conditions (one-way ANOVA, $F = 5.41$, $p = 0.001$); the range of recalled memories in terms of graph distance is controllable within the range of $0$ to $5$. In Appendix A.4, we show the correlation matrices for all patterns.

Figure 1: Correlations of memory patterns within 10-hop neighbourhoods of the triggered memory pattern's vertex in $M = C_{30}$. The $k$-hop neighbourhood is the set of vertices within a distance of $k$ edges from the triggered memory pattern. For $a, h$ pair, all vertices ($n = 30$) are tested, and here we plot the mean $\pm$ standard deviation.

## 3.2 MULTI-SCALE REPRESENTATIONS OF COMMUNITY STRUCTURES IN GRAPHS

Now we will consider more interesting memory graph topologies. *Zachary's karate club graph* Zachary (1977) consists of $34$ vertices, representing karate practitioners, where edges connect individuals who consistently interacted in extra-karate contexts. Notably, the club split into two halves. Setting Zachary's karate club graph as $M$ and varying $a$ and $h$, however, reveals that there were even finer social groupings than these, as Figure 2 reveals and as we discuss in Appendix A.5.

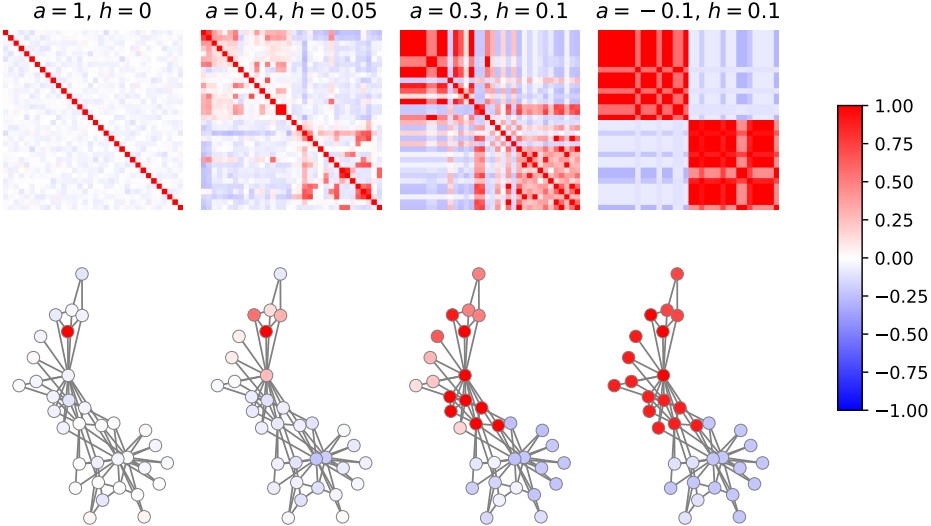

Figure 2: Memory pattern correlations for each vertex in $M$, set as Zachary's karate club graph. The top row shows correlations between each pair of attractors, where colour indicates the correlation coefficient, $1$ (red) to $-1$ (blue). The bottom row draws $M$ with vertices coloured by the correlation coefficients at $S^{(101)}$, which is dynamically stable (see Appendix A.5).

To more clearly illustrate the multi-scale representations of graph communities, we also test CDAM on the *barbell graph* (see Appendix A.6 for further details) and the *Tutte graph* (see Figure 3 and Appendix A.1).

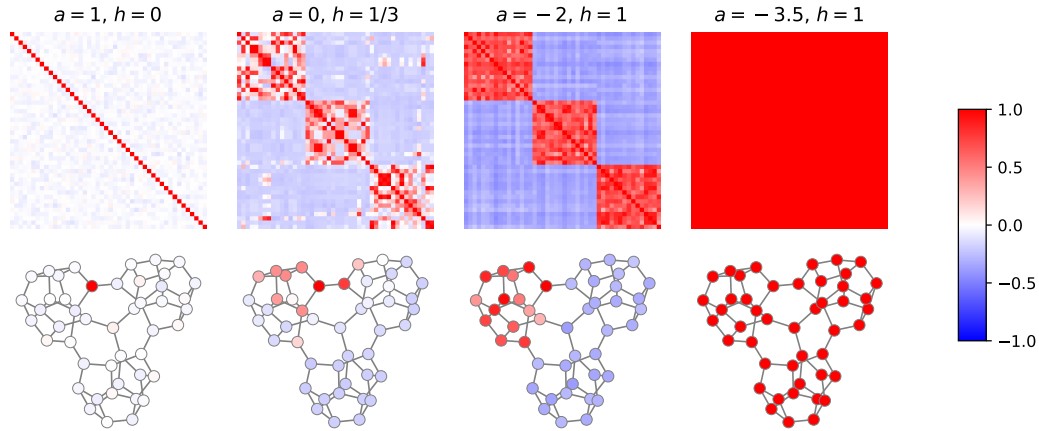

Figure 3: Correlations between the convergent meta-stable states ($S^{(101)}$ values from Figure 7) for all pairs of trigger stimuli (top row); and $M$ drawn with vertices coloured by these meta-stable state correlations for a particular trigger stimulus (bottom row).

### 3.3 SPARSE TEMPORAL SEQUENCE RECALL OF REAL VIDEO DATA

Hetero-association is naturally suited for encoding temporal sequences. Here we use a directed cycle graph $\overrightarrow{C_{50}}$ where the patterns are sparsely sampled frames of videos (see Appendix A.7 for details).

Figure 4 shows activity over time in a network with $M = \overrightarrow{C_{50}}$. At each step $t$ of the simulation, we calculate the correlation of $S^{(t)}$ with each pattern. We start the simulation by triggering the first pattern (frame) and thereafter leave the network to continue its dynamics according to Equation 1. Importantly, we require sufficient Anti-Hebbian auto-association, i.e., $a < 0$, in combination with relatively strong Hebbian hetero-associations, i.e., $h > 0$. Otherwise, the sequence recall can become stuck or lags due to auto-correlations.

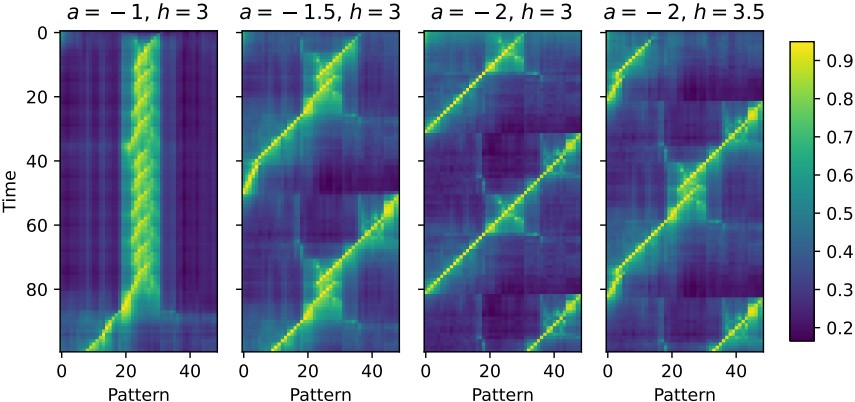

Figure 4: Correlations of memory patterns over time for each vertex in $M = \overrightarrow{C_{50}}$, where each memory pattern is a sparsely sampled video frame (see Appendix A.7 for details) from video 1.

Notably, similar settings for Anti-Hebbian auto-association and Hebbian hetero-association is required for a different sparsely sampled video, as shown in Figure 5. Only in the case of $a = -2, h = 3$ can we recall the sequence without skips or delays. Present in both Figures 4 and 5, we

can see more global features in the video and sharp context switches. These structures can be also be seen in the correlations between the attractors (see Appendix A.7).

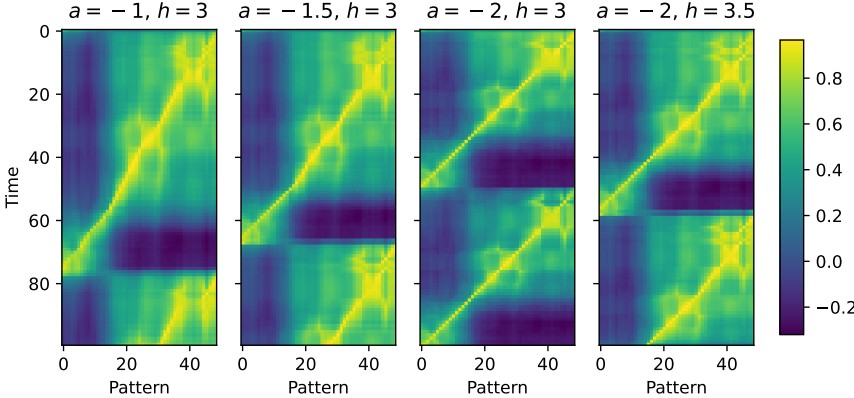

Figure 5: Correlations of memory patterns over time for each vertex in $M = \overrightarrow{C_{50}}$, where each memory pattern is a sparsely sampled video frame (see Appendix A.7 for details) from video 2.

## 4   CONCLUSION

In this paper we have introduced a new dense associative memory model, called *Correlated Dense Associative Memory (CDAM)*, which auto- and hetero-associates continuous-valued memory patterns using an underlying (arbitrary) graph structure. Using such memory graph structures, and especially by modulating recall using anti-Hebbian auto- or hetero-association, we demonstrated extraction of multiple scales of representation of the community structures present in the underlying graphs. We additionally tested CDAM with perhaps the most traditional and obvious application of hetero-associative memory networks – temporal sequence memory – with sparsely sampled real-world videos. Here, the benefits of anti-Hebbian modulations were highlighted once again, this time in its role as a stabiliser against internal correlations (natural distractors) within a sequence and of ordered recall generally.

### 4.1   IMPLICATIONS AND FUTURE WORK

For neuroscience, this work highlights the highly non-trivial contributions of anti-Hebbian learning to the proper functioning across a range of tasks, including controlling the sequence recall range, community detection in graphically-organised memories, and temporal sequence retrieval. These findings invite experimentalists to further explore the contribution of inhibitory neurons in cognition.

For machine learning, perhaps one of the most impactful uses of this work will be in its application to improving the performance and/or understanding of Transformer models (Vaswani et al., 2017) through their connection to continuously-valued dense associative memory networks (Krotov & Hopfield, 2016; Ramsauer et al., 2021). Indeed, Ramsauer et al. (2021) used this connection to study the 'attractive schemas' of the implied energy landscape in a large language model. This generated hypotheses about the function of particular layers and attention heads in the model, and may potentially help us further elucidate the internal representational structure of similar models. As Millidge et al. (2022) notes, Transformers' attention mechanism can be interpreted in its mathematical form as performing hetero-association between its keys and and values in the associative memory sense. Can we use this insight to identify the topology of the attractor or energy landscape in models trained on language, image recognition, or other tasks? Do such models entrain particular structures such as memory graphs (or higher dimensional analogues) to reflect the topology of the underlying data structures and correlations within the training set? And could a modulatory mechanism such as an anti-Hebbian learning rule help direct the 'flow' of temporally-evolving cognition, such as in-context or one-shot learning in large language models (Brown et al., 2020)? These and many other questions are now open for exploration, and will hopefully offer us deeper insights into the inner-workings of some of the most performant and powerful ML systems used today.

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

# A  APPENDIX

## A.1  NUMERICAL SIMULATIONS EXHIBITING FOUR DYNAMICAL MODES & E–I BALANCE

In the numerical simulations below, we use $N = 1,000$, $\beta = 1$, and $\eta = 0.1$. We choose values of $a$ and $h$ to demonstrate the four dynamical modes described in Subsection 2.2. Simulations are terminated at $t = 101$, which in all cases is a fixed point or limit cycle. The memory patterns stored are random vectors as described in Subsection 2.1.

To initialise the network state in each simulation, we choose a memory pattern $\mu \in M$ and set $S^{(0)} = \xi^\mu + cX$, where $X$ is a random vector with elements independently drawn from the interval $[-0.5, 0.5]$ and $c \in \mathbb{R}^+$ is the amplitude of the additive random noise. Here we use $c = 1$.

Figure 6 shows results for a 2–regular graph (the only type of which are unions of 1D cycles). We can notice some apparent 'clusters' of vertices which appear to commonly become co-active. This represents a common meta-stable state shared by the surrounding trigger stimuli. The reason for these particular meta-stable groups is due to the random biases present in the random patterns, which likely become amplified by a finite field effect. Notably, in the wide hetero-association condition ($a = -1, h = 1$), the meta-stable groups are fewer in number and larger in size than the narrow hetero-association condition ($a = 1, h = 1$).

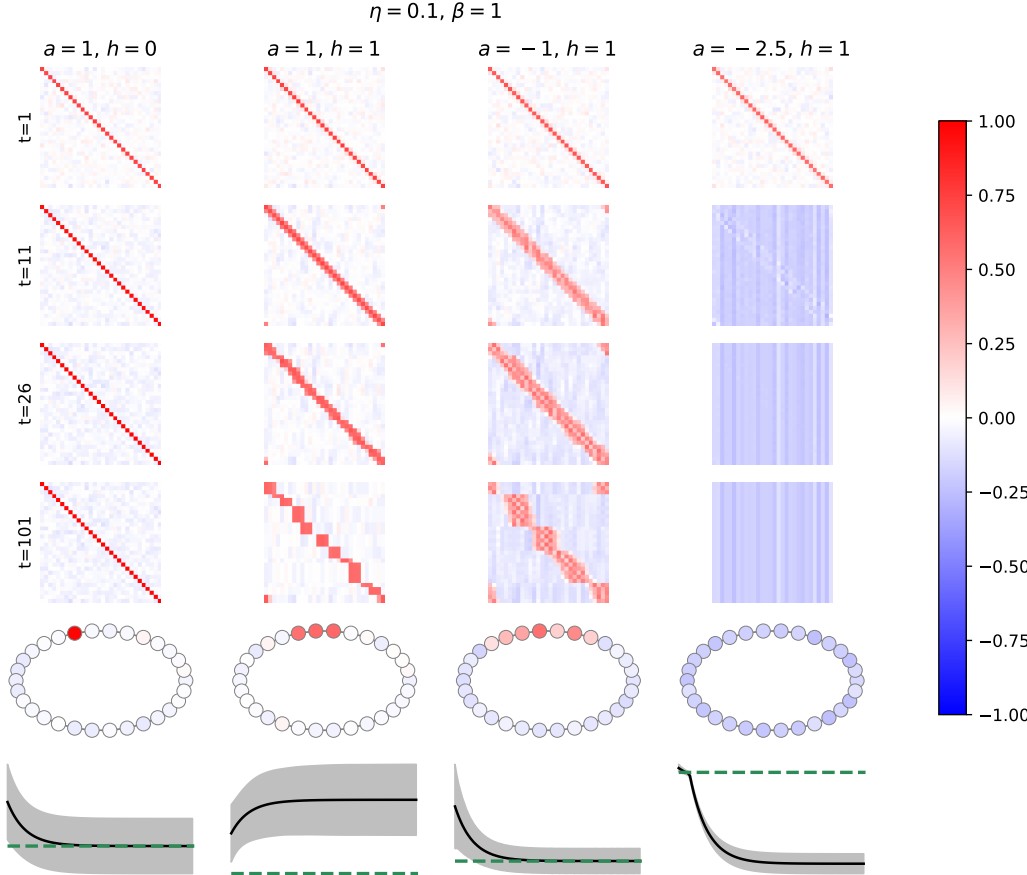

Figure 6: Setting $M$ as an undirected cycle graph of length $P = 30$, we demonstrate the four dynamical modes of the network (from left to right): auto-association, narrow hetero-association, wide hetero-association, and neutral quiescence. At $t = 1, 11, 26, 101$, we plot the correlation between each memory pattern and the current state, $r(\mu^{(t)})$. In the penultimate row, we draw $M$ with vertices coloured by $r(\mu^{(t)})$ for one initial trigger stimulus. And in the final row, we plot the mean $\pm$ standard deviation of the neural activities over time, with the dotted green line at 0.

Figure 7 shows results for the Tutte graph (Tutte, 1946), which is 3–regular. Unlike the 1D cycle graph shown in 6, the Tutte graph has a more interesting topology, in the form of 3 clusters of highly-connected vertices. These clusters become noticeable by looking at the emergent structure of the correlations for the wide hetero-association case ($a = -1, h = 1$). However, it becomes even more noticeable by looking at the correlations between the convergent meta-stable states, which we show in Figure 3.

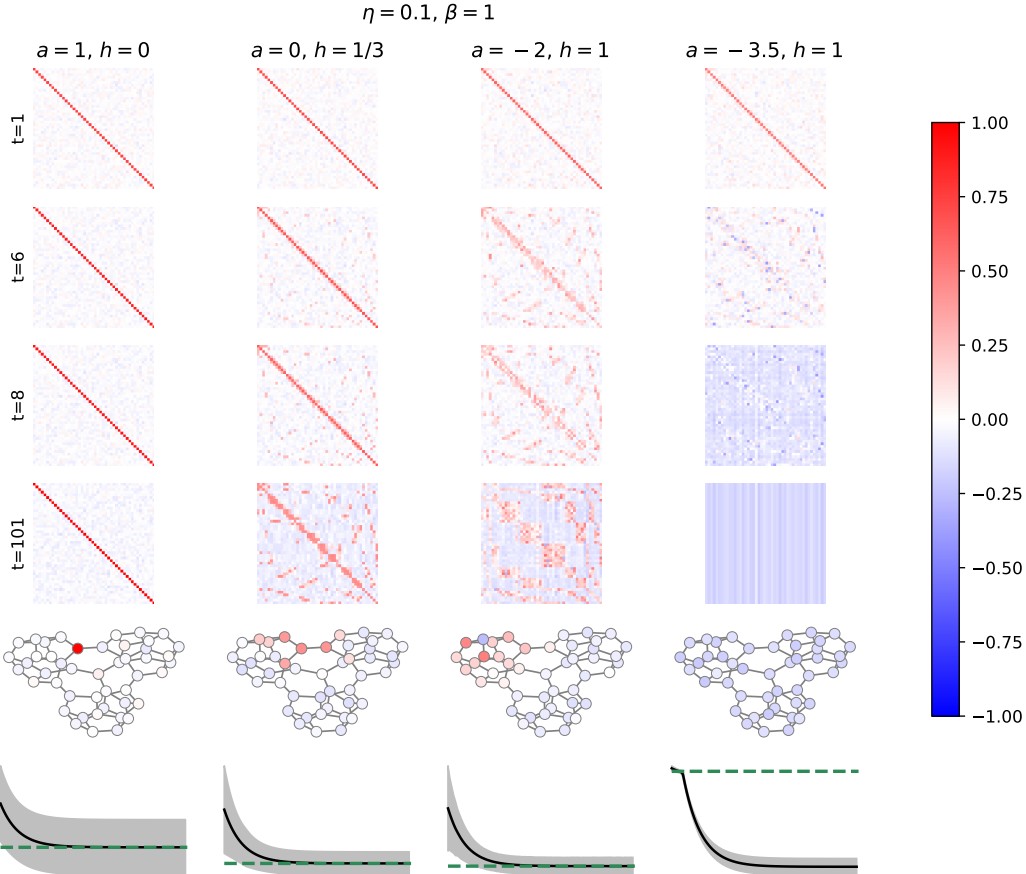

Figure 7: Setting $M$ as the Tutte graph (Tutte, 1946) ($P = 46$), we demonstrate the four dynamical modes of the network (from left to right): auto-association, narrow hetero-association, wide hetero-association, and neutral quiescence. At $t = 1, 11, 26, 101$, we plot the correlation between each memory pattern and the current state, $r(\mu^{(t)})$. In the penultimate row, we draw $M$ with vertices coloured by $r(\mu^{(t)})$ for one initial trigger stimulus. And in the final row, we plot the mean $\pm$ standard deviation of the neural activities over time, with the dotted green line at 0.

Finally, in Figure 8 we analyse a random 3–regular graph where $h = \frac{1-a}{m}$ and $a = 1, 0.5, 0, -1$. In each case, we can see the network converges to an unbiased E–I balance (bottom row of Figure 8). We can also see that as $a$ decreases in value, the spread of hetero-association becomes gradually wider. However, unlike in the Tutte graph, there are no natural clusters of vertices. Therefore, the resulting correlation matrices reflect the random topology insofar as having no discernible regularity, besides the approximately uniform distribution of noise (which is uniform due to the regular nature of the graph, causing each trigger stimulus to activate a similar number of other memory patterns).

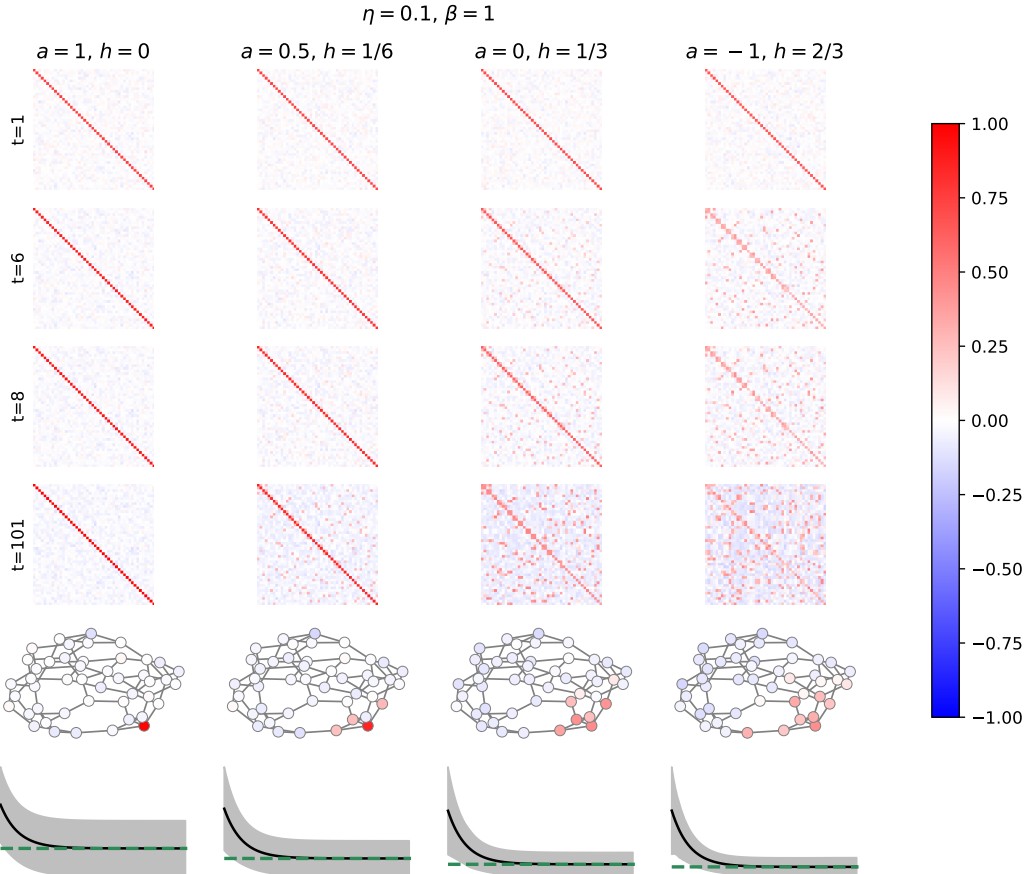

Figure 8: Setting $M$ as a random 3–regular graph with $P = 46$, we demonstrate the four dynamical modes of the network (from left to right): auto-association, narrow hetero-association, wide hetero-association, and neutral quiescence. At $t = 1, 11, 26, 101$, we plot the correlation between each memory pattern and the current state, $r(\mu^{(t)})$. In the penultimate row, we draw $M$ with vertices coloured by $r(\mu^{(t)})$ for one initial trigger stimulus. And in the final row, we plot the mean $\pm$ standard deviation of the neural activities over time, with the dotted green line at $0$.

## A.2 1D CYCLE MEMORY GRAPHS

Practically all of the past semantic hetero-associative literature (Amari, 1972; Tank & Hopfield, 1987; Kleinfeld & Sompolinsky, 1988; Gutfreund & Mezard, 1988; Griniasty et al., 1993; Gillett et al., 2020; Tyulmankov et al., 2021; Karuvally et al., 2023; Chaudhry et al., 2023; Karuvally et al., 2023) has studied the case of $M$ being a 1D cycle. This is because such models typically consider $P$ memory patterns, and construct weights between neurons $i$ and $j$ in a form such as

$$J_{ij} = \frac{1}{N} \sum_{\mu}^{P} (\xi_i^{\mu} \xi_j^{\mu} + \xi_i^{\mu+1} \xi_j^{\mu} + \xi_i^{\mu} \xi_j^{\mu+1}), \tag{6}$$

or

$$J_{ij} = \frac{1}{N} \sum_{\mu}^{P} (\xi_i^{\mu+1} \xi_j^{\mu} + \xi_i^{\mu} \xi_j^{\mu+1}), \tag{7}$$

where, crucially, the memory patterns are semantically correlated along a single line. This would make $M$ a line graph, where it not for the fact that most studies let $\xi^{P+1} = \xi^1$, which connects the two ends of the line to form a circle, as shown in Figure 9.

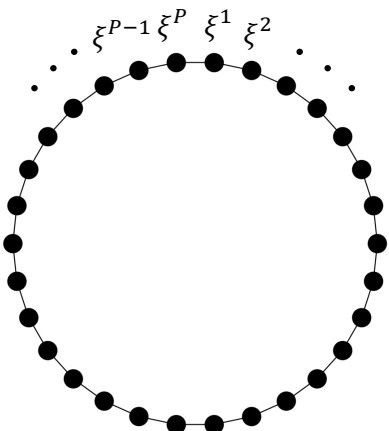

Figure 9: Illustration of the 1D cycle memory graph, with the vertices labelled by the memory index. Note that this would be a line if we do not identify $\xi^{P+1} = \xi^1$.

In other appendices and the main text, we denote a 1D cycle graph with $n$ vertices as $C_n$.

### A.3 Replication of Miyashita (1988)

As described in Subsection 1.2, Miyashita (1988) is a a classical study in the semantic hetero-association neuroscience literature, which showed neurons from monkey temporal cortex were responsive to the presentation of stimuli according to the order in which they were presented. These semantic links were developed without general regard to any spatial or statistical similarities shared by the stimuli. In neurons which were significantly responsive to the stimuli, their activity was significantly auto-correlated with the activity elicited by the stimuli up to a distance of 6 patterns into the future.

We manually transcribed data from Figure 3C of Miyashita (1988) by printing the enlarged figure and carefully using a pencil and ruler to measure data for the 28 cell group (illustrated as square symbols in the original figure), which showed the largest hetero-associations. The mean and standard error of the mean (SEM) which we measured and used in the subsequent analysis are shown in Table 1.

Table 1: Auto-correlations between neural activities responsive to visual stimuli in the monkey temporal cortex. The data are transcribed from the 28 cell group (square symbols) of Figure 3C in Miyashita (1988). Distance refers to the temporal distance between the stimuli.

| Distance | 0 | 1 | 2 | 3 | 4 | 5 | 6 |
|---|---|---|---|---|---|---|---|
| Mean | 1 | 0.33810 | 0.19700 | 0.11940 | 0.08806 | 0.07015 | 0.06493 |
| SEM | 0 | 0.03731 | 0.03582 | 0.02985 | 0.02388 | 0.02015 | 0.02239 |

Here we model the results of Miyashita (1988) by setting $M = C_{30}$ (see Appendix A.2) and choosing $a$ and $h$ to match the data. As shown in Figure 10 We find that an anti-Hebbian auto-association and Hebbian hetero-association ($a = -1.3, h = 1$) correlated well with the experimental results.

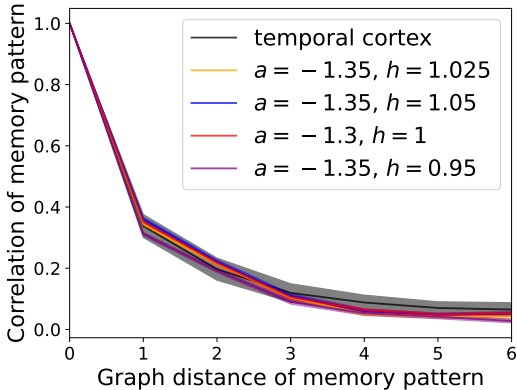

Figure 10: Mean $\pm$ standard error of the means for $M = C_{30}$ with different values of $a$ and $h$, alongside the transcribed Miyashita (1988) data shown Table 1. The closest matching model tested was $a = -1.35, h = 1.05$, which had an $R^2 = 0.996$ with the Miyashita (1988) data.

### A.4 CONTROLLING THE RANGE OF ATTRACTORS

Figure 11 shows the case of $M = C_{30}$ with varying levels of $a$ and $h$ and random memory patterns (and draws its data from the same simulations as for Figure 1). At $a = 1$ and $h = 0$, we have the expected auto-associative behaviour. However, as we decrease $a$ and set $h = \frac{1-a}{2}$ (to maintain unbiased E–I balance), we see an increase in hetero-association and a gradually increasing spread of excitation through the graph, with excitation emanating from the triggered memory pattern that was set at $S^{(0)}$.

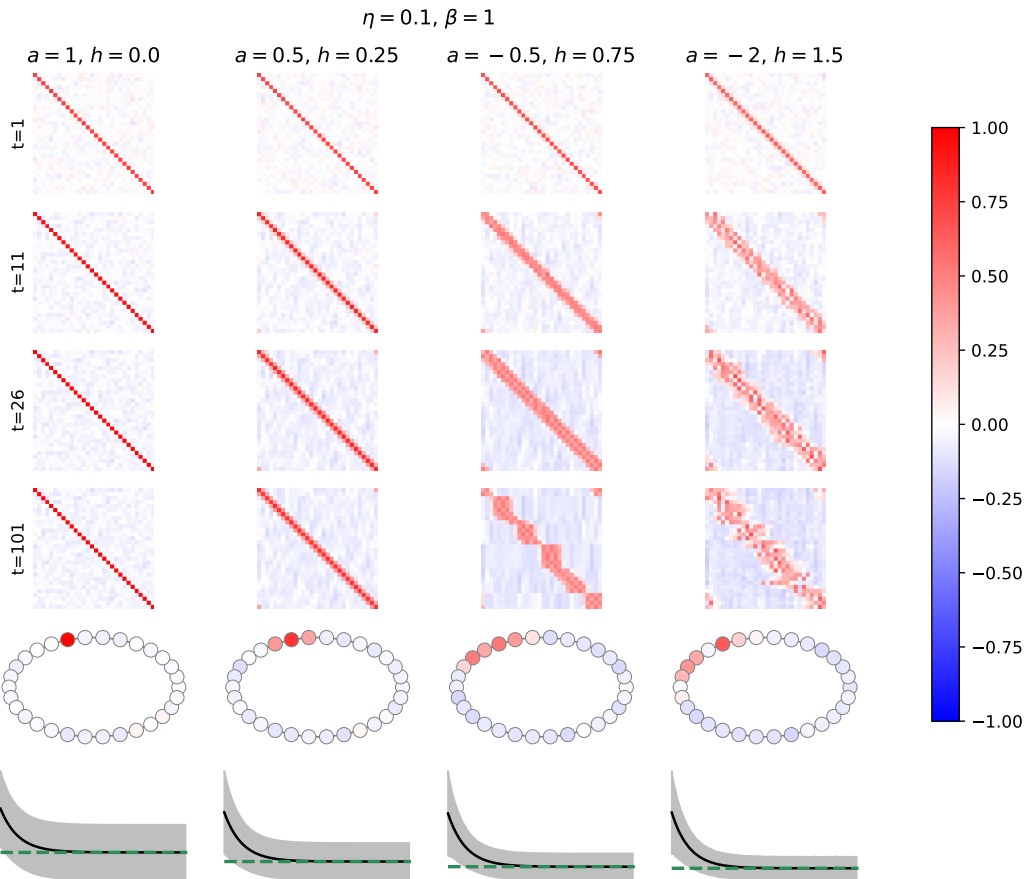

Figure 11: Memory pattern correlations for each vertex in $M = C_{30}$ with increasing range of hetero-association (left column to right column). The first four rows show the correlations as a heatmap of dimensions $30 \times 30$, where each cell is coloured by its correlation coefficient, $1$ (red) to $-1$ (blue). The penultimate row draws $C_{30}$ with vertices coloured by the correlations at the end of the simulation for the same trigger stimulus (the vertex coloured red in the left-most column) for each tested set of parameters.

## A.5 ZACHARY'S KARATE CLUB GRAPH

An interesting and naturally-constructed graph is *Zachary's karate club graph* Zachary (1977). It consists of $34$ vertices, representing karate practitioners, where edges connect individuals who consistently interacted in extra-karate contexts. Notably, the club split into two halves. Setting Zachary's karate club graph as $M$ and varying $a$ and $h$, however, reveals that there were even finer social groupings than these, as seen in Figures 2 and 12. Smaller groups are particularly noticeable in some of the individual pattern trigger stimuli for $a = 0.4, h = 0.05$ (Figure 14) and $a = 0.3, h = 0.1$ (Figure 15). Contrastingly, $a = 1, h = 0$ selects for individuals (Figure 13) and $a = -0.1, h = 0.1$ selects for the two major groups post-split (Figure 16).

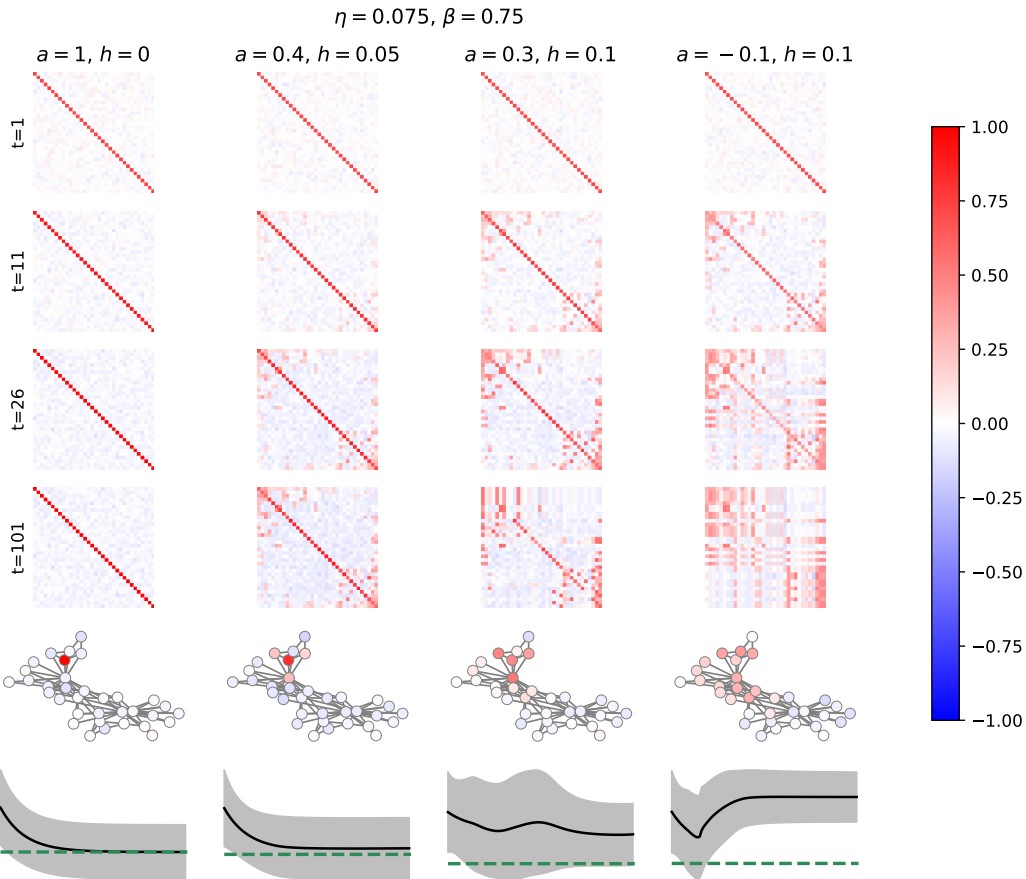

Figure 12: Setting $M$ as Zachary's karate club graph (Zachary, 1977), we demonstrate multi-scale graph segmentation. At $t = 1, 11, 26, 101$, we plot the correlation between each memory pattern and the current state, $r(\mu^{(t)})$. In the penultimate row, we draw $M$ with vertices coloured by $r(\mu^{(t)})$ for one initial trigger stimulus. And in the final row, we plot the mean $\pm$ standard deviation of the neural activities over time, with the dotted green line at $0$, demonstrating dynamic stability.

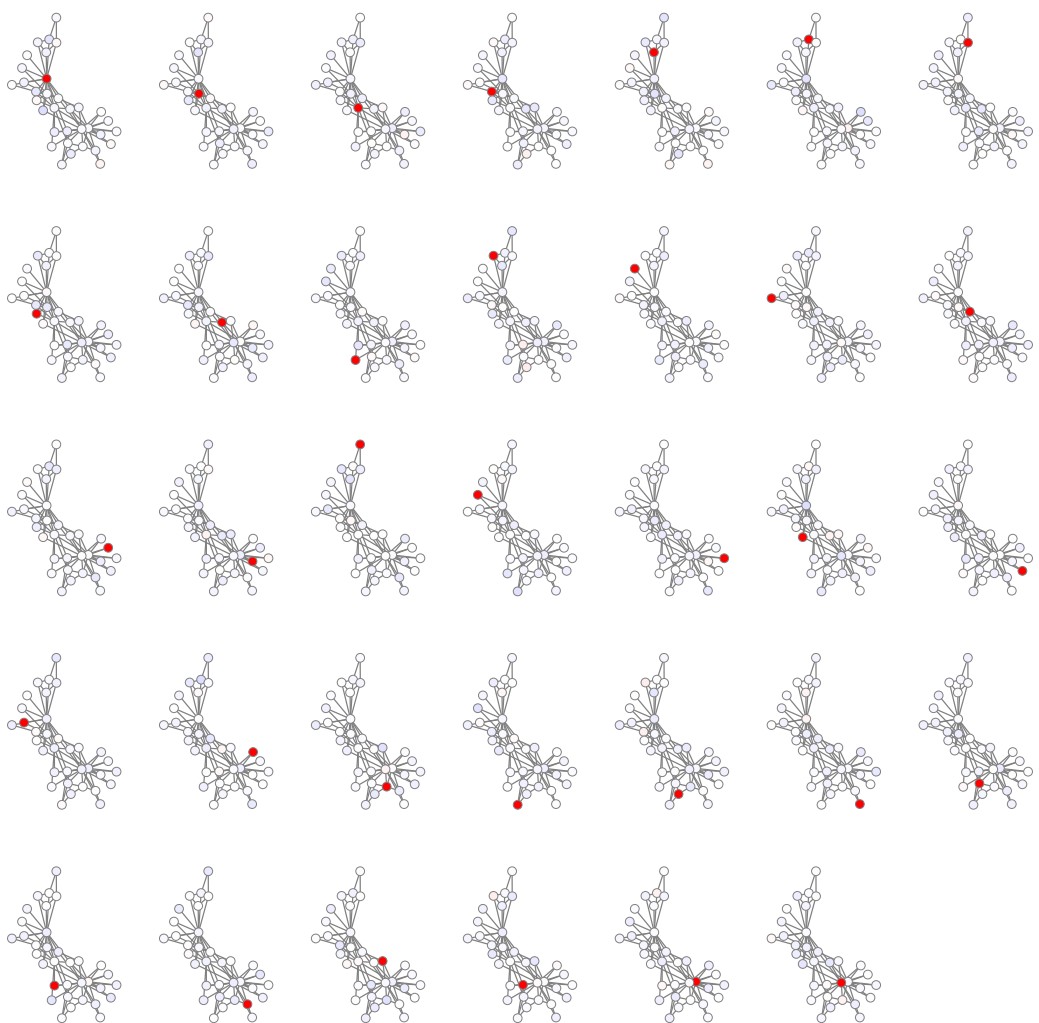

Figure 13: $S^{(101)}$ correlations given different every possible trigger stimulus in Zachary's karate club graph with $a = 1$ and $h = 0$.

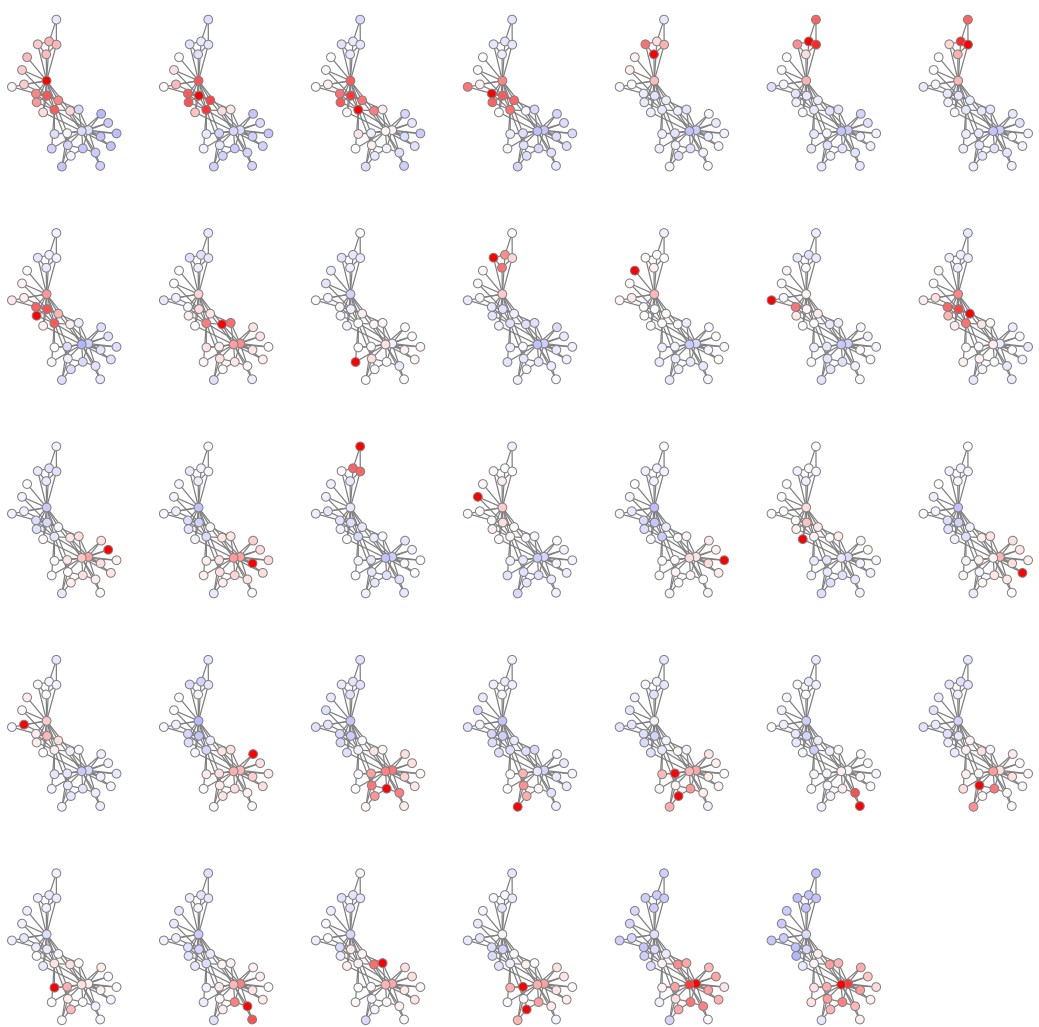

Figure 14: $S^{(101)}$ correlations given different every possible trigger stimulus in Zachary's karate club graph with $a = 0.4$ and $h = 0.05$.

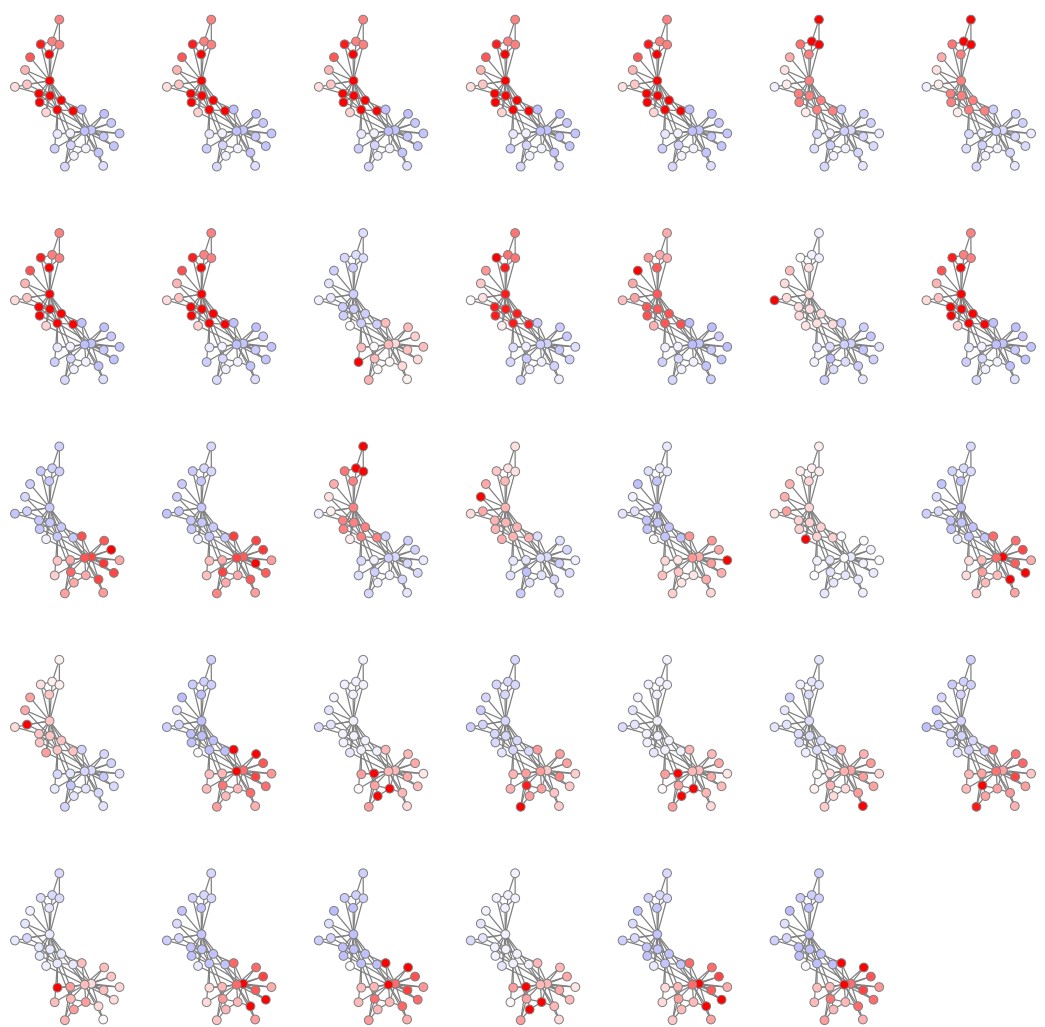

Figure 15: $S^{(101)}$ correlations given different every possible trigger stimulus in Zachary's karate club graph with $a = 0.3$ and $h = 0.1$.

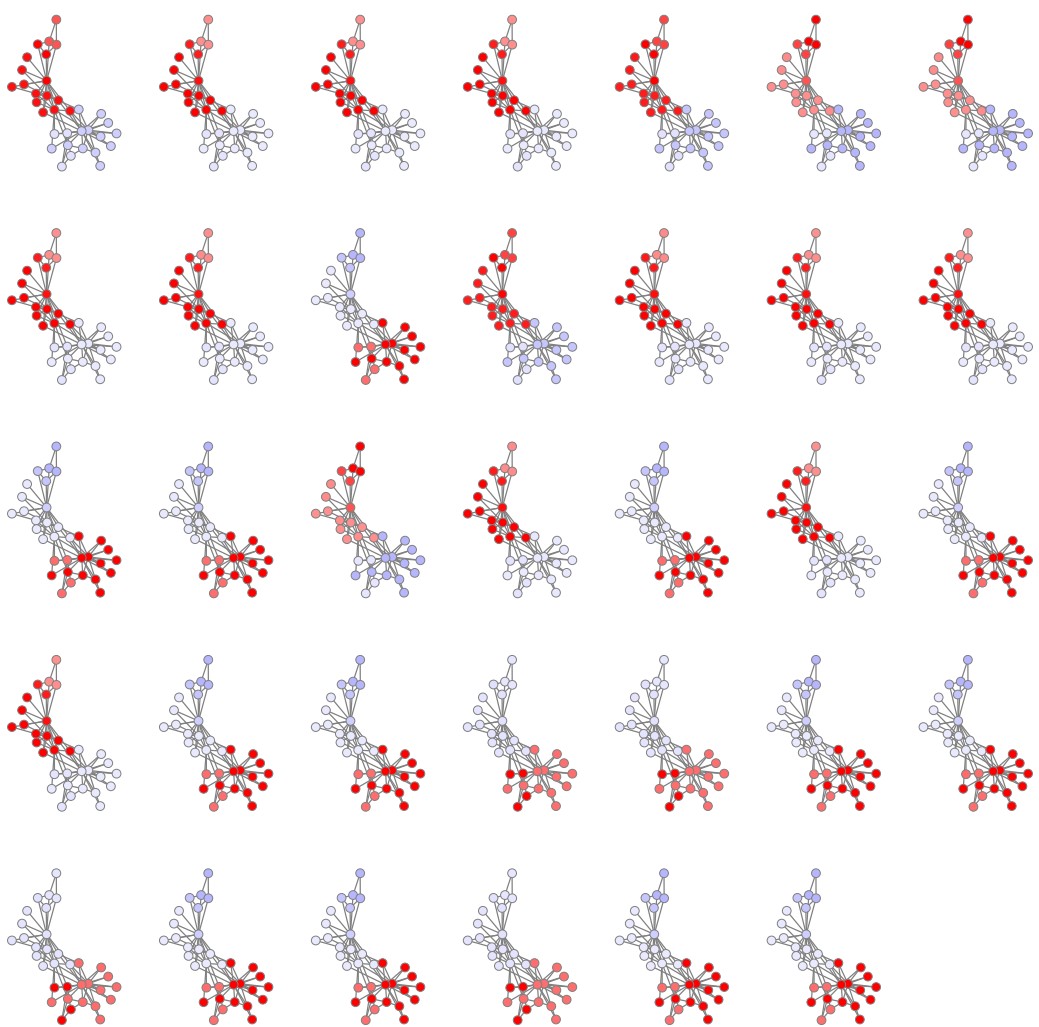

Figure 16: $S^{(101)}$ correlations given different every possible trigger stimulus in Zachary's karate club graph with $a = -0.1$ and $h = 0.1$.

### A.6 BARBELL MEMORY GRAPH

A *barbell graph* $B_{n,m}$ is the union of two copies of the complete (fully-connected) graph $K_n$ on $n$ vertices, connected by a single path vertices of size $m$. Here we choose $n = m = 10$. This simple example helps to demonstrate the two extremes of the attractive regime scales – where one scale maintains individual pattern activities and the other identifies the local pattern cliques in $M$.

Figure 17 shows correlations for random memory patterns embedded in $M = B_{10,10}$ with varying levels of $a$ and $h$. At $a = 1$ and $h = 0$, there is no hetero-associative activity, only auto-association. However, as we decrease $a$ with fixed $h$ the two $K_n$ cliques quickly show correlated group activity. Along the path connecting the two complete graphs, we also see a lengthening in the spread of activity along the path (like in the cycle graph). This can be further verified by inspection of the individual attractors, where Figures 18–21 show $S^{(101)} \cdot \xi^\mu$ values for each trigger stimulus pattern in the graph across tested values of $a$ and $h$.

It is notable that of the tested conditions, only $a = 1, h = 0$ converges to stable fixed points, whereas the other tested conditions demonstrate chaotic attractors and epileptiform activity with increasing amplitude as $a \to 0$. This is likely caused by a finite field effect which recurrently amplifies a small, noise-induced bias in the memory patterns.

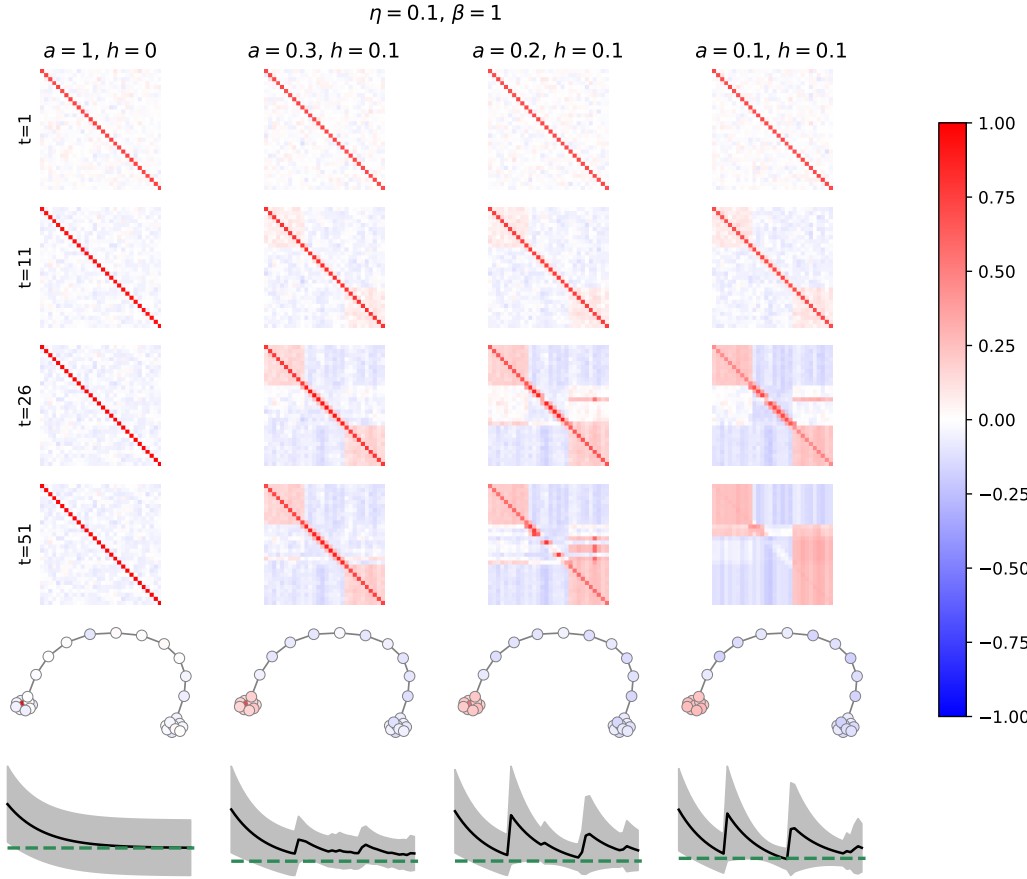

Figure 17: Memory pattern correlations for each vertex in $M = B_{10,10}$ with decreasing $a$ (left column to right column). The top row shows the correlations as a heatmap of dimensions $30 \times 30$, where each cell is coloured by its correlation coefficient, $1$ (red) to $-1$ (blue). The bottom row shows an example of the mean terminal activity states given the same pattern trigger stimulus (the vertex coloured red in the left column) for each tested pair of $a$ and $h$ values.

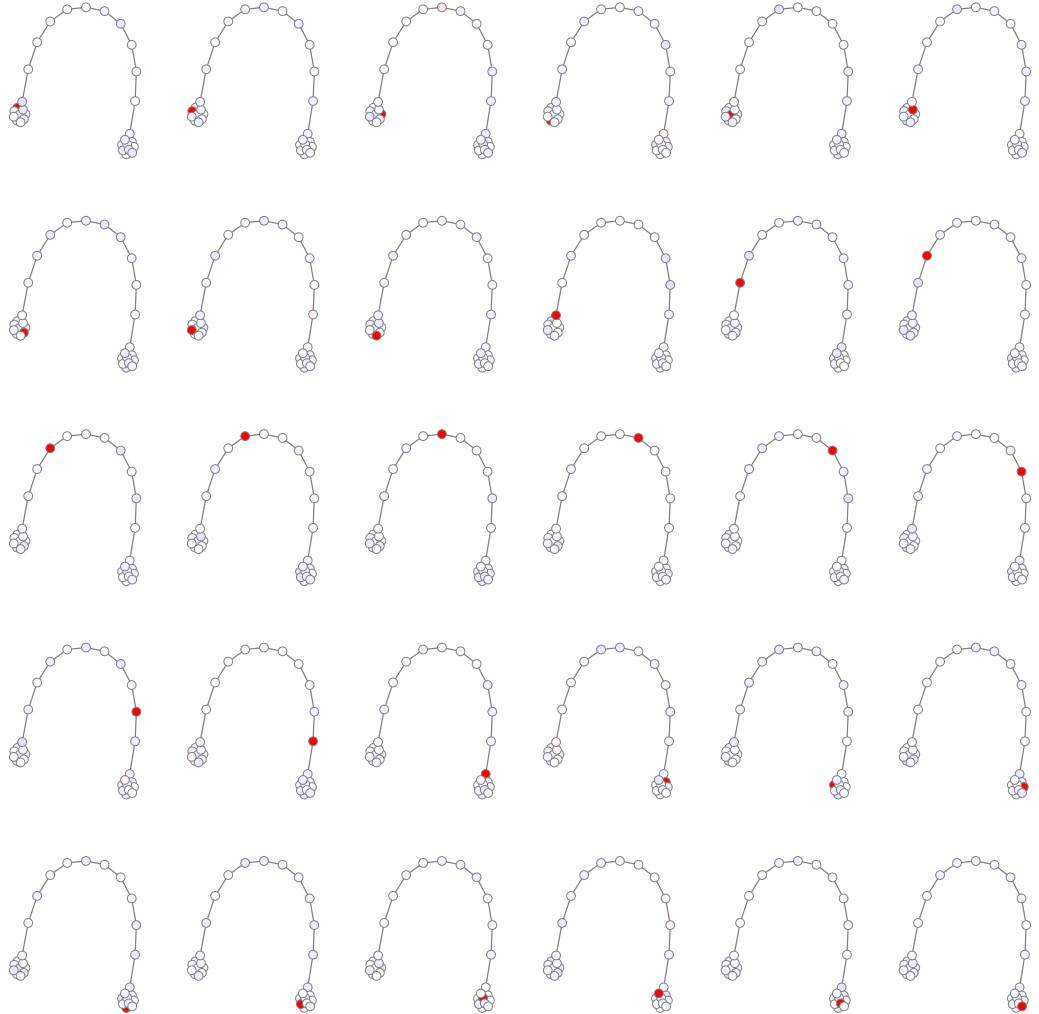

Figure 18: $S^{(101)}$ values given different every possible trigger stimulus in $B_{10,10}$ with $a = 1$ and $h = 0$.

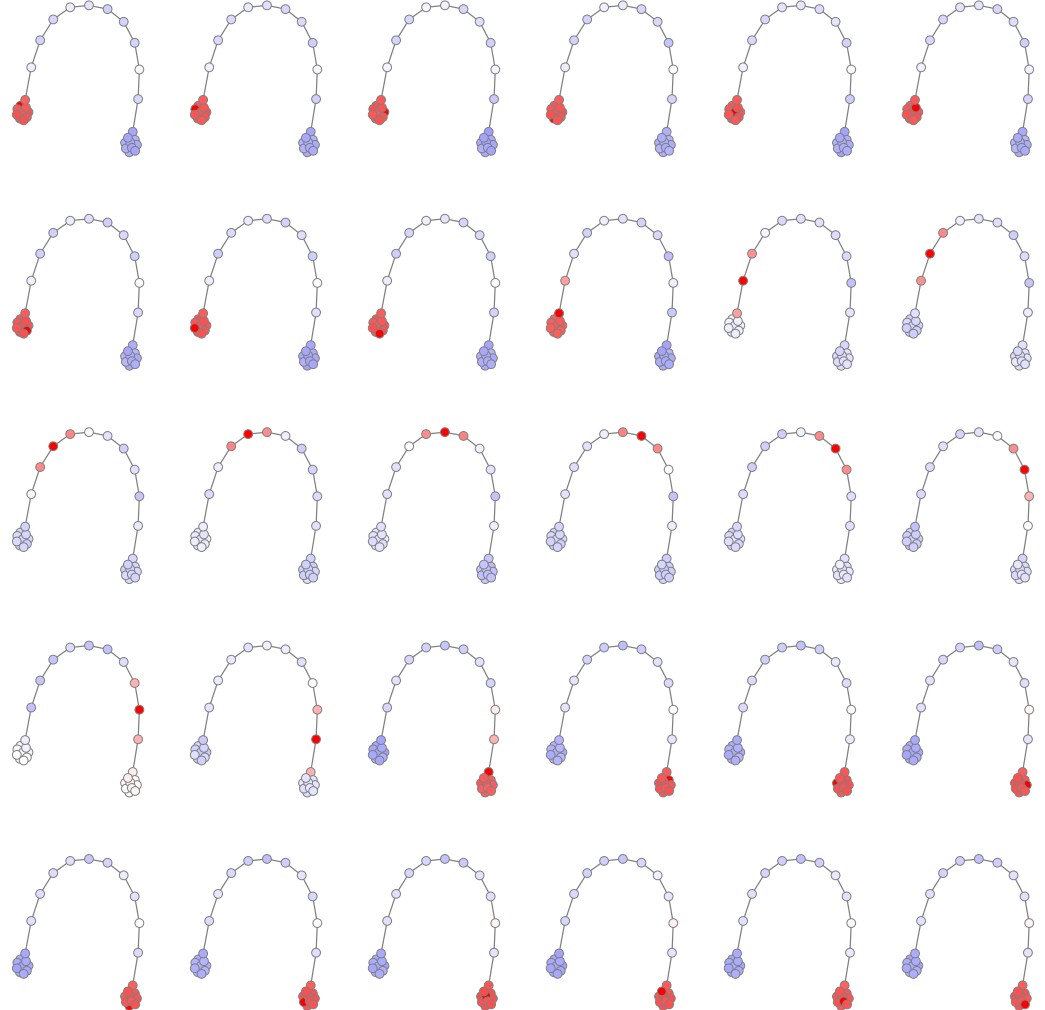

Figure 19: $S^{(101)}$ values given different every possible trigger stimulus in $B_{10,10}$ with $a = 0.3$ and $h = 0.1$.

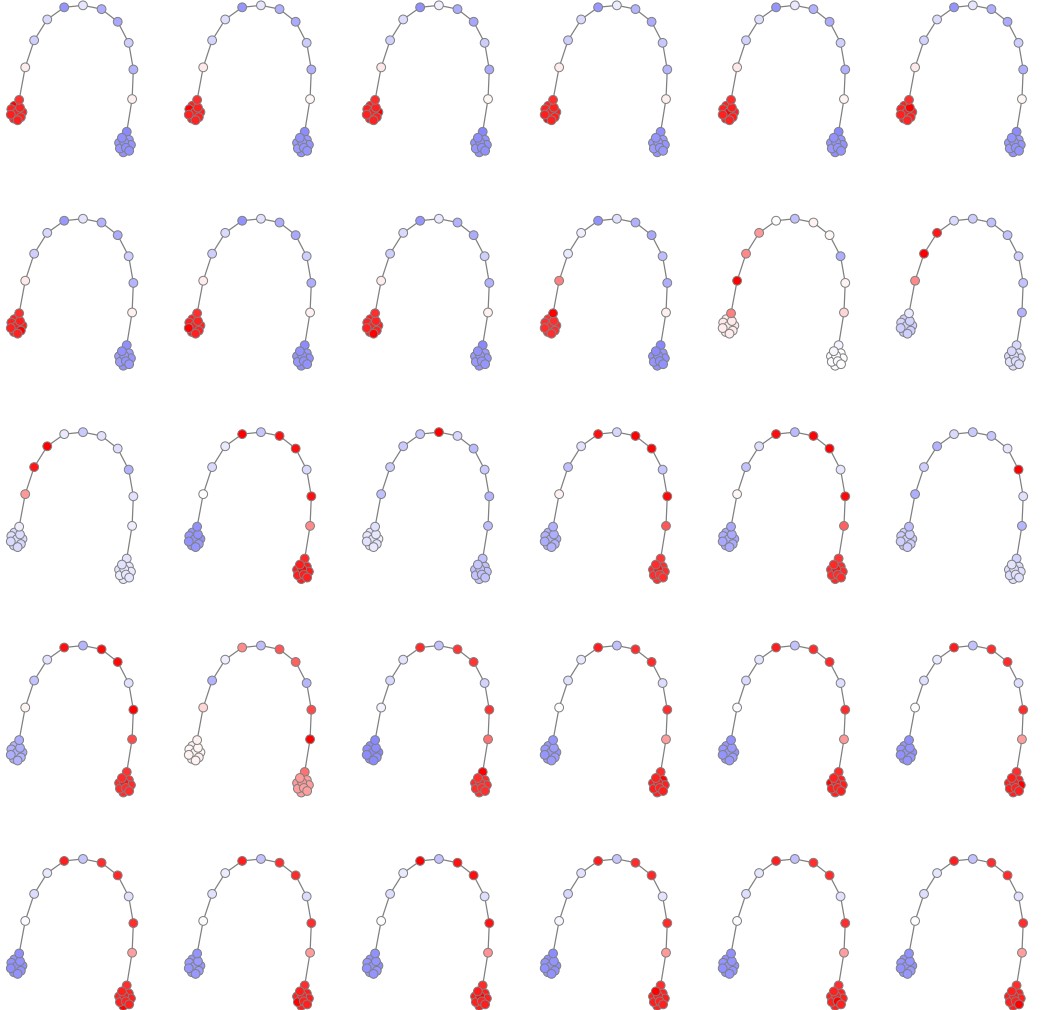

Figure 20: $S^{(101)}$ values given different every possible trigger stimulus in $B_{10,10}$ with $a = 0.2$ and $h = 0.1$.

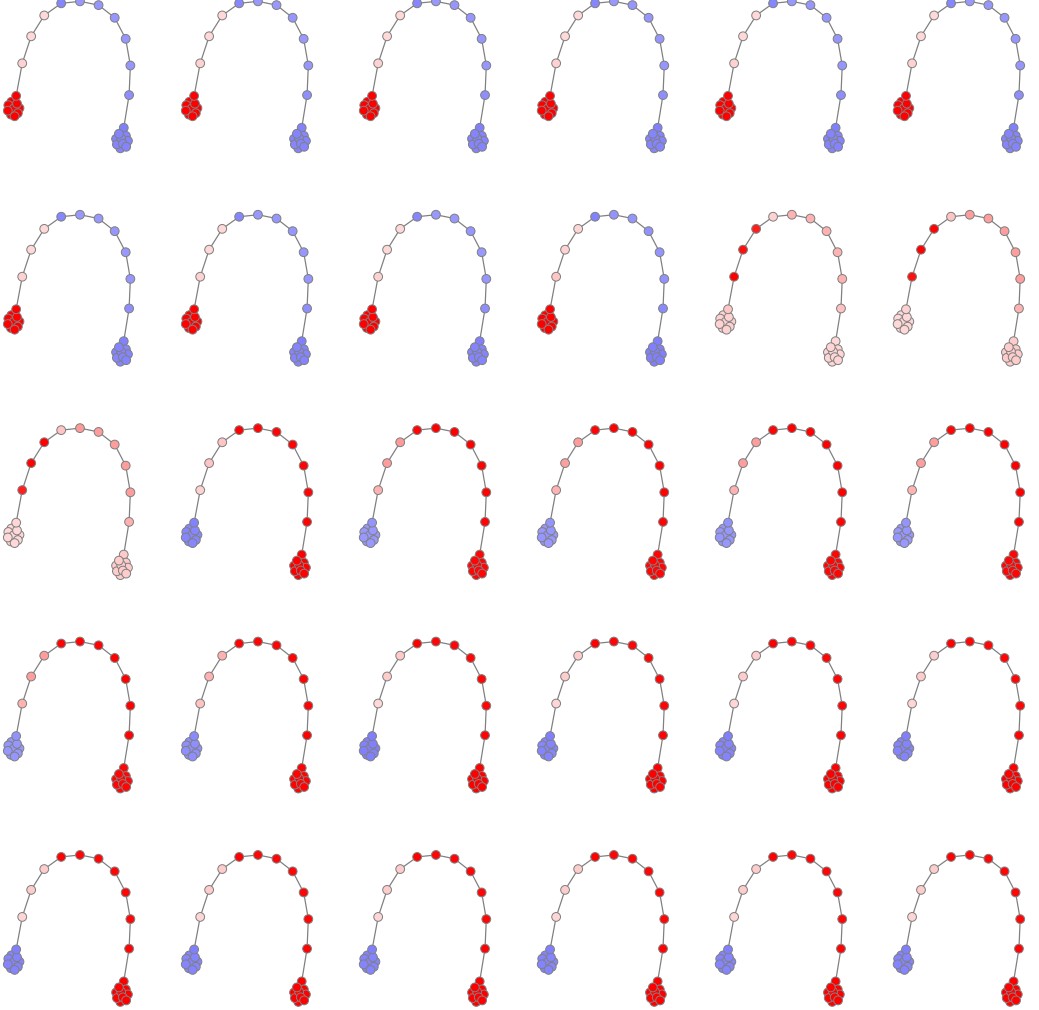

Figure 21: $S^{(101)}$ values given different every possible trigger stimulus in $B_{10,10}$ with $a = 0.1$ and $h = 0.1$.

### A.7 VIDEO SEQUENCE RECALL

The two videos used were sourced from Wikimedia Commons and were uploaded by User:Raul654 on 24 January 2006. They can found at the below URLs:

`https://commons.wikimedia.org/wiki/File:Gorilla_gorilla_gorilla1.ogv`

`https://commons.wikimedia.org/wiki/File:Gorilla_gorilla_gorilla4.ogv`

We used the first 50 frames of both videos were used. The videos are size 320px $\times$ 240px with bit depth of $24$. The pixel and colour information was flattened into a single vector of length $230,400$, with the values normalised by the maximum value, $240$. We then randomly sample $N = 2,000$ values from the image and treat these as our neural patterns and states. For illustration, the first frames of each video are shown in Figure 22.

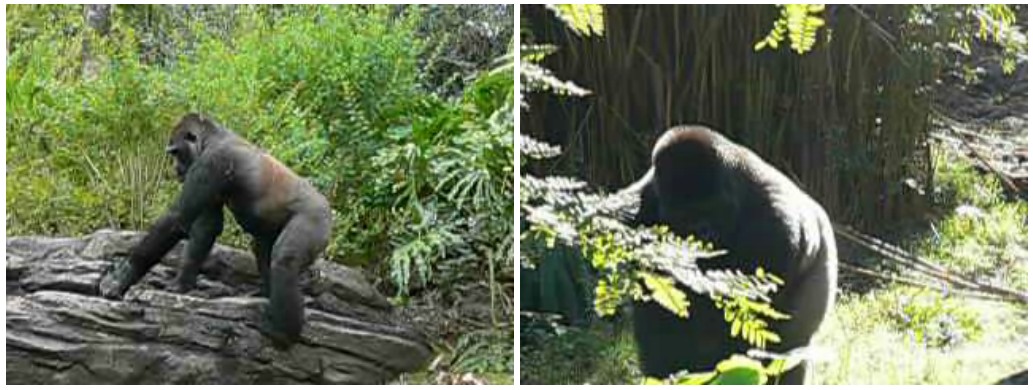

Figure 22: First frames of the two videos used in the video recall experiment: video 1 (left) and video 2 (right).

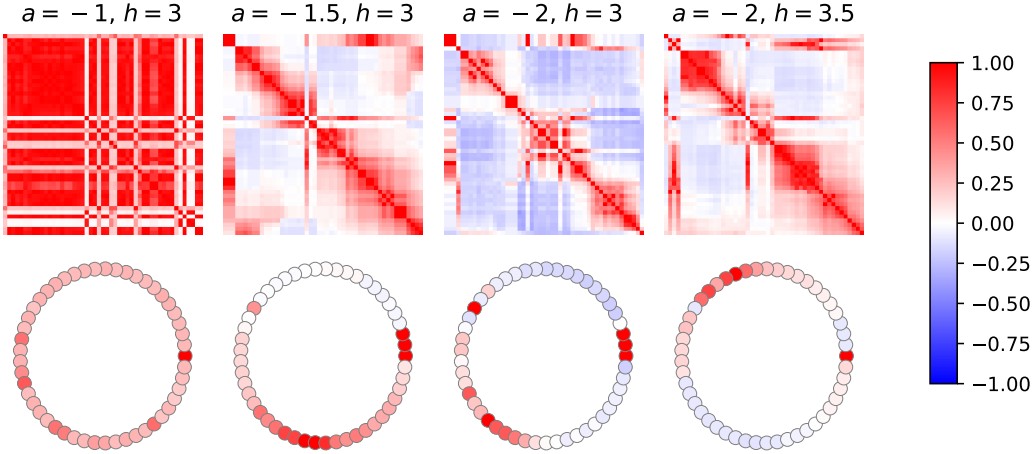

Figure 23: Correlations between attractors for each target stimuli of video 1.

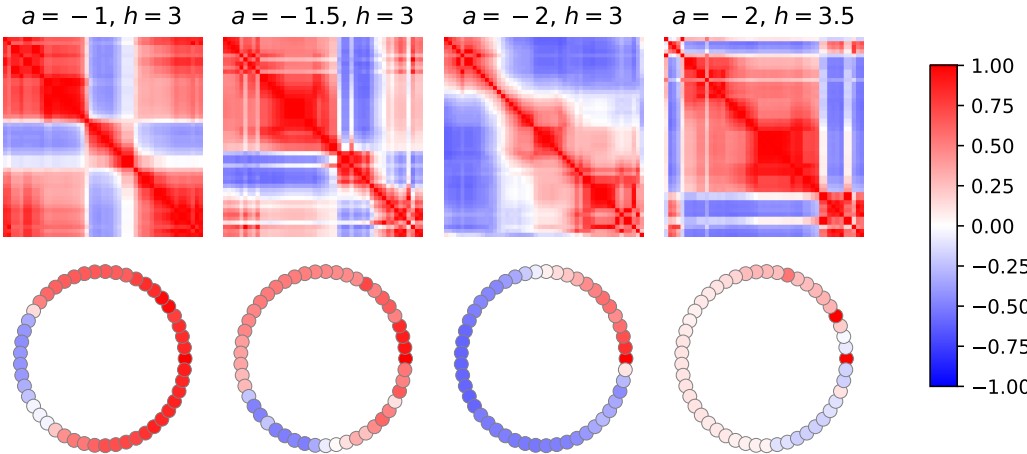

Figure 24: Correlations between attractors for each target stimuli of video 2.

