# OpenReview forum: "Correlated dense associative memories"
_ICLR.cc/2024/Conference — Submitted to ICLR 2024_

### Official Review · Reviewer_oKZA · 2023-10-18

**Soundness:** 2 fair
**Presentation:** 2 fair
**Contribution:** 2 fair
**Rating:** 3
**Confidence:** 4

**Summary:**

This paper proposes a model for heteroassociative memory that extends previously-proposed sequence memory models to store more general heteroassociations.

**Strengths:**

There is at present much interest in extensions of the classic Hopfield network, and the links between those extensions and the transformer architectures that are now ubiquitous in deep learning. This paper is therefore timely, and of potential interest to the community working on such memory models.

**Weaknesses:**

In the Abstract, the authors claim that "[their] results have implications for both machine learning and neurobiology." However, after reading the paper I was left wondering what precisely those implications were. In my **Questions** below, I detail several concerns regarding the presentation and substance of the results.

**Questions:**

1. The overall motivation of the paper is, to me, rather unclear. Yes, the paper extends past work on sequence modeling using heteroassociative memories, but it is not made clear why this extension is of interest. This lack of clarity is accentuated by the fact that the simulations are not woven together into a coherent story. Rather, they read as independent vignettes, and it is not made clear why each is scientifically interesting.

2. In the list of contributions, the authors make an interesting claim regarding links between their model and studies of inhibitory plasticity in neuroscience. However, this link is never made concrete later in the paper. If there is a deeper link to neuroscience experiment, expanding on this point could help resolve my concerns regarding overall motivation, framing, and significance.

3. The weighted combination of autoassociative and heteroassociative terms in the update rule (2) resembles that used in previous works on discrete-state sequence models, c.f. the cited work of Kleinfeld & Sompolinsky (1988), work by Kanter & Sompolinsky (1986, Temporal Association in Asymmetric Neural Networks, not cited in the submitted manuscript), or eq. 13 of the cited work by Chaudhry et al. (2023). In those sequence models, tuning the relative weighting of the associative and heterassociative terms changes the dwell time of the system in a given state. Here, no clear motivation for the inclusion of the autoassociative term is given. The authors' empirical results (see for instance Figure 2) are consistent with the intuition that networks with small $b$ dwell near the initial state for longer, but they do not probe this carefully. Indeed, I think the authors' results on how varying $b$ determines which scales are "selected" are more precisely explained in terms of an increase in dwell time. This would be easy to probe by examining intermediate timesteps. In particular, they should examine how their model's dynamics relate to diffusion on the graph. This is also relevant to the question of what the overall objective of the proposed heteroassociative learning rule is. Finally, the parameterized weighting in (2), with $b$ and $1-b$, is a bit puzzling given that the authors proceed to consider $b = -100$; at this scale they are effectively considering $b$ and $-b$ as the weights. In short, more specific motivation for the combined update rule should be provided (including for why the authors are interested in continuous memories in this particular setting), and the resulting behavior probed more carefully.

4. In a similar vein to the above, the simulations do not probe some of the most basic properties of associative memories. How does the capacity of their model scale with the number of neurons? Related to point (3) above, how quickly does the model converge to each target state?

---

> ### Author Response · Authors · 2023-11-23
> **Reply to Reviewer oKZA**
>
> We greatly appreciate Reviewer oKZA’s guidance and direction in the form of their detailed review of our paper. In the points below, we wish to respond to these with details of how we have improved the paper based on their feedback.
>
> *Weaknesses*
>
> Reviewer oKZA found that the implications for machine learning and neurobiology were unclear. In conjunction with additional experiments, we have added to the Introduction and Discussion to highlight our contributions.
>
> Questions:
>
> 1. Thank you for the feedback. We have now overhauled major parts of the paper to more clearly communicate our results and tell a coherent, overarching story.
>
> 2. `In the list of contributions, the authors make an interesting claim regarding links between their model and studies of inhibitory plasticity in neuroscience. However, this link is never made concrete later in the paper. If there is a deeper link to neuroscience experiment, expanding on this point could help resolve my concerns regarding overall motivation, framing, and significance.`
>
> To make this more concrete, we discuss our motivations from neuroscience in the introduction and replicate data from a neuroscience experiment.
>
> 3. `no clear motivation for the inclusion of the autoassociative term is given`
>
> We have expanded on our motivations from both neuroscience and machine learning in the introduction.
>
> Our initial inclination was to construct our model such that we could rigorously test it while varying a single parameter, $b$. However, as the reviewers noted, this became less natural outside the range of $b \in [0,1]$. So, while parsimony of this kind has its advantages, we completely agree with the reviewers that it offers little advantage in this case. We therefore separated the auto- and hetero-associative strengths into two parameters – $a$ for the auto-associative weighting and $h$ for the hetero-associative weighting. This reparameterization further aided us in the theoretical analysis of the model.
>
> 4. `How does the capacity of their model scale with the number of neurons? [And] how quickly does the model converge to each target state?`
>
> When $a, h \neq 0$, regular notions of ‘capacity’ no longer seem fitting. This is because ‘capacity’ is normally measured in the pure auto-associative case by giving a noise-corrupted or partial memory pattern, and observing whether and how closely the model's dynamics converge to the uncorrupted or complete memory pattern (e.g., see Amit et al. 1985 for the classical model and Demircigil et al. 2017 for the dense model). In the pure hetero-associative case, ‘capacity’ has (to our knowledge) only ever been studied in the linear sequences case (e.g., see Löwe 1998 for the classical model and Chaudhry et al. 2023 for the dense model). However, in our model we study general mixtures of both auto- and hetero-association, as well as arbitrary memory graphs (not just linear cycles). It is therefore unclear whether there exists an appropriate notion of ‘capacity’ for our model.
>
> One can, however, study the model in a similar spirit of analysis. To this end, we have completed theoretical and numerical work to demonstrate the dynamics of our model in the thermodynamic limit. By analysing the energy function, we show how (under certain assumptions of the memory graph) the network can exhibit one of four dynamical modes – auto-association, narrow hetero-association, wide hetero-association, and neutral quiescence – as controlled by choices of $a$ and $h$. Additionally, we provide some propositions on how parameter choices and the memory graph structure influences dynamics.

---

### Official Review · Reviewer_TfVu · 2023-10-30

**Soundness:** 2 fair
**Presentation:** 3 good
**Contribution:** 2 fair
**Rating:** 3
**Confidence:** 4

**Summary:**

The authors introduce a new associative memory model, which they name Correlated Dense Associative Memories (CDAM). Much of the prior work on associative memory models focus on the case of auto-associative memory, in which networks learn to recover memory patterns given parts of those same memory patterns. The authors instead explore hetero-associative memory retrieval, in which the recovered memory patterns differ from the given patterns. Using random and partially-random data, the authors analyze the properties of CDAM networks. They further study different network topologies and their impact on the embedded associative memory structures, and show that it can be used to model multi-scale representations of community structures in graphs, oriented recall in a symmetric connection regime, and temporal sequences with distractors.

**Strengths:**

- The introduction provides a good overview of the field. It gives a crisp summary of the intuition behind associative memory networks, and delineates the field’s current progress and some open problems.

**Weaknesses:**

- The introduction barely discusses the contributions of this work; the contributions are relegated to single-sentences at the end with few details. It’s hard to interpret the results or impact of this work from the intro-- much of it reads like a background.
- The results section is hard to parse; it’s unclear to me what research questions the authors are trying to answer with their results section. I suggest adding an overview section and explicitly stating the goals of your experiments.
- Experimental design is poorly discussed. It's difficult to find basic information about the experiments, such as what dataset is used, or what refutable hypotheses each experiment is testing. The authors mention that they used both random and partially-random memory patterns; in the partially-random case they "use some real-world data." Where does this real-world data come from and what are its structural properties?
- There is insufficient discussion of the potential applications or usefulness of CDAM. The authors state that “our results have implications for both machine learning and neurobiology” but do not convincingly back up this claim. They discuss some connections to transformers in the future work section, but it is purely speculative.
- Frankly, this work seems very rushed and premature. It's less than 8 pages, with nearly half of the content occupied by figures. Brevity itself is certainly not bad, but when there are so many components of the paper inadequately discussed or simply missing, a short submission reflects poorly on the authors. Submitting incomplete work unfairly appropriates reviewer resources from complete submissions. I strongly discourage the authors from doing so again in the future, lest they risk tarnishing their academic reputation.

**Questions:**

- The novelty claim of this work is not immediately clear to me. Can you explain how CDAM qualitatively differs from the method and capabilities of hetero-associative MCHNs discussed by Millidge et al. (2022) which you cite?
- The experiments are mostly run on randomly generated data. This makes its impact on real-world problems hard to judge. How does CDAM perform on real-world data, or how do you expect it to perform?
- On what types of real-world problems should I use CDAM, and why? You claim that your results show that CDAM can be used to model multi-scale representations of community structures in graphs, oriented recall in a symmetric connection regime, and temporal sequences with distractors. In which domains do these problems appear, and how does CDAM compare to current approaches?
- Could you explain what anti-Hebbian learning rule you applied and why?

---

> ### Author Response · Authors · 2023-11-23
> **Reply to Reviewer TfVu (part 1)**
>
> We are very thankful for Reviewer TfVu’s patience in instructing us on ways to improve our paper. We have worked to integrate all feedback we received, and in our ‘General Response’ summarise the changes and additions made. Below we address specific points raised by Reviewer TfVu.
>
> *Weaknesses* (ordered by the corresponding dot-points from Reviewer TfVu’s initial review):
>
> 1. We appreciate Reviewer TfVu’s praise of the introduction (under ‘Strengths’) insofar as it provided a `good overview` and `crisp summary` of the field, its intuitions, progress, and open problems. Nevertheless, we recognise it is helpful if the introduction also incorporates details about the results and impact of the current work. We have therefore revised the introduction section to integrate further details of this paper’s contributions, including two new sub-sections (one for neuroscience, one for machine learning).
>
> 2. We thank Reviewer TfVu for the suggestion; we have added an overview subsection to the beginning of the results section, where we explicitly state the goals of each experiment.
>
> 3. In our initial submission, no real datasets were used – all data were simulated*. In our original submission (in the final paragraph of section 3.3, titled ‘Oriented recall in a symmetric regime’) we said that if a modeller wishes to dedicate some of the memory pattern vector contents (e.g., half) to intentionally create correlations between some patterns, this would reduce the amount of usable memory (a fact reflected in the reduced capacity of classical associative memory networks, e.g., see Theorem 2.2 of Löwe, Annals of Applied Probability 1998). In the case of random patterns (as we used in that experiment), this matters little since we are only interested in the dynamics of the network, not the actual contents of the random memories. It was in this context that we stated `The downside of this technique is that it reduces the usable content of memory vectors, e.g., where instead of using random memory patterns we use some real-world data.` However, the wording `we use` is confusing, so we have edited this sentence to clarify that we used random memory patterns in that experiment, not real-world data.
>
> *Based on suggestions from reviewers, in our revision we include new experiments with real data.
>
> 4. Reviewer TfVu comments that we did not adequately discuss potential applications or uses of CDAM. Therefore, we have added further discussion to address this, as well as added new theoretical and numerical analyses to demonstrate potential applications vis-à-vis including graph clustering and replicating a neuroscience experiment’s results. We have also further added to the Discussion section regarding the link to Transformers, as well as a subsection addressing neuroscience implications.
>
> 5. While we submitted this paper in good faith, we certainly acknowledge there were gaps (as the reviews have pointed out). We greatly appreciate all reviewers’ assistance in helping improve the quality of our work. Additionally, for full context, the reason why the original submission’s main content was ~7.5 pages was because we mistakenly believed the initial submission had a page limit of 8 pages (rather than the correct number, 9), and we left approximately half a page for the author list/affiliations. We regret not checking this more closely, and hope the reviewers will recognise our efforts in improving our work in response to their feedback.
>
> *Questions* (ordered by the corresponding dot-points from Reviewer TfVu’s initial review):
>
> 1. `The novelty claim of this work is not immediately clear to me. Can you explain how CDAM qualitatively differs from the method and capabilities of hetero-associative MCHNs discussed by Millidge et al. (2022) which you cite?`
>
> In Millidge et al. (2022), section 3.5 describes how we may generally consider the relationships between auto- and hetero-associative models, and notes how Transformers’ attention mechanisms take the hetero-associative form. Appendix B of Millidge et al. (2022) goes on to use the simple example of splitting CIFAR10 and MNIST images into upper and lower halves, treating the upper-half as the query and the lower-half as the hetero-associated memory (or ‘value’, in the Transformer language). So, given the upper-half of a CIFAR10 or MNIST image, the network can recall the lower-half. This is where the work of Millidge et al. (2022) stops.

---

> ### Author Response · Authors · 2023-11-23
> **Reply to Reviewer TfVu (part 2)**
>
> In the CDAM framework, we can consider the hetero-associative experiments shown in Millidge et al. (2022)’s Appendix B as one which sets the memory graph very specifically as a union of all memorised pairs of upper- and lower-half memories (assigned to two separate vertices) of CIFAR10 or MNIST images (where the the upper-half vertex has a directed edge to the lower-half vertex). In the current paper, we work with arbitrary memory graphs and demonstrate a much more general framework, with applications and connections to, e.g., graph clustering and replicating data from a neuroscience experiment.
>
> 2. & 3. Reviewer TfVu asks about performance on real-world data and tasks.
>
> We have now included new experiments which use real-world data. We hasten to add that applying associative memory models directly to real-world problems is normally inadvisable where domain-specific models exist. It is common, however, for researchers to marry existing, e.g., gradient-based, models with associative memory models, e.g., Saha et al. ‘End-to-end Differentiable Clustering with Associative Memories’ ICML 2023 and Auer et al. ‘Conformal Prediction for Time Series with Modern Hopfield Networks’ NeurIPS 2023. Such integrations constitute independent works in their own right, and are beyond the scope of this paper, which aims to investigate associative memory networks at a more fundamental level and from a more theoretical perspective, e.g., similar to Chaudhry et al. ‘Long Sequence Hopfield Memory’ NeurIPS 2023 or Burns & Fukai ‘Simplicial Hopfield networks’ ICLR 2023. Nevertheless, we acknowledge that it is helpful for applied researchers to see examples of potential applications, thus our additions.
>
> More broadly, we wish to emphasise one of the theoretical aims of this paper is to provide evidence of the inherent computational capabilities of dense hetero-associative networks. This is scientifically interesting from the perspective of AI interpretability due to the connection between CDAM and the attention mechanism of Transformers. Our results suggest that single attention heads in Transformers could be performing any or all of the demonstrated computations, including simulating computations using combinations of auto- and hetero-association. This constitutes a notable addition to the recent literature on Transformers’ capabilities in this area, e.g., see Liu et al. ‘Transformers Learn Shortcuts to Automata’ ICLR 2023.
>
> 4. `Could you explain what anti-Hebbian learning rule you applied and why?`
>
> We used a symmetric rate-based rule, since this is a rate-based model without a notion of spike timing (for background, see Roberts & Leen, 2010 and Shulz & Feldman, 2013). In our initial parametrization of the model with $b$, when $b<0$ the model entered an anti-Hebbian auto-associative and Hebbian hetero-associative regime. Our initial inclination was to construct our model such that we could rigorously test it while varying this single parameter. However, as reviewers indicated (and we agreed), this is slightly unnatural given the structure of the model’s dynamics (Eq 2 in the initial submission, Eq 1 in the revision). We therefore separated the auto- and hetero-associative strengths into two parameters – $a$ for the auto-associative weighting and $h$ for the hetero-associative weighting. This reparametrization aids in interpretation of the model and avoids the unnaturalness present in the prior use of highly negative values of $b$. It also allows for any combination of anti-Hebbian and Hebbian rules applied to the auto- and hetero-association (with anti-Hebbian corresponding to negative values of $a$ or $h$, whereas Hebbian corresponds to positive values).

---

> ### Comment · Reviewer_TfVu · 2023-12-05
>
> Thank you for the detailed response.
> While I certainly appreciate the improvements, I maintain my position that the original submission was very premature.
> It does not look like a simple misunderstanding of the page requirements, which is what the authors are claiming.
> This is reflected in the number of changes made– the updates to this paper are so significant that it is almost a new paper deserving a new review entirely.
> Abusing the pre-rebuttal period to complete an incomplete submission should be discouraged: this is not fair to other papers, nor to the scientific community.
> Reviewers will either have to take attention away from other papers to re-review this work, or rush their review within the rebuttal period, making their reviews less trustworthy.
>
> That said, I do believe the work has greatly improved.
> In particular, I found that the intro better presents the motivations of this work, and the objectives of the experiments in the Numerical Simulations sections are much clearer; I also appreciate the inclusion of the experiments on real-world video data.
>
> However, I still have some significant remaining concerns:
>
> - I don't believe that the authors adequately addressed my concerns regarding the practical applications of CDAM.
> While the new experiments show that CDAM may be used to encode frames of videos, the author do not clearly explain why learning this encoding is useful (for example, does this help with solving any downstream tasks? Can it be used to replace the encoder part of a transformer to improve performance?), nor compare it to other existing methods of sequence encoding.
> Furthermore, the experiments are evaluated on a very small dataset, using 50 frames of 2 videos, which does not provide enough evidence to support the generalizability of this approach.
> - The authors' claimed implications of this work for ML are still very hand-wavy, stating that "perhaps one of the most impactful uses of this work will be in its application to improving the performance and/or understanding of Transformer models."
> This is too speculative for this work's results to be considered a significant contribution for ML.
> - The implications of this work to neuroscience also appear inadequate.
> The significance of inhibitory neurons in cognition is already well-established (O'Reilly et al., 2012).
> It is also a rather large leap to argue that the importance of anti-Hebbian learning in CDAM is suggestive of the importance of anti-Hebbian learning in biological cognition.
> To make such a claim, the authors would need stronger evidence to show that CDAM closely replicates the biological process of memory formation, or that CDAM can be used as a computational model that explains memory formation while satisfying all known biological constraints. Demonstrating that the behavior of CDAM correlates with that of Miyashita (1988) is a step in this direction, but not enough by itself.
>
> Therefore, I think this work, as currently presented, has neither sufficiently significant implications for ML nor neuroscience.
> I am increasing my score to a 3 because it has improved from the original manuscript, but I still recommend reject.
>
>
> **References**
>
> [1] O'Reilly, Randall C., et al. Computational cognitive neuroscience. Vol. 1124. Mainz: PediaPress, 2012.
>
> [2] Miyashita, Yasushi. "Neuronal correlate of visual associative long-term memory in the primate temporal cortex." Nature 335.6193 (1988): 817-820.

---

### Official Review · Reviewer_K2Ug · 2023-10-31

**Soundness:** 3 good
**Presentation:** 2 fair
**Contribution:** 3 good
**Rating:** 6
**Confidence:** 4

**Summary:**

The paper proposes a generalization of recently studied static and sequential Dense Associative Memories to settings where the sequences of memories live on a graph with connectivity matrix H, instead of living on a one dimensional line. The proposed update rule is also related to hetero-associative memory settings, which is a valid perspective.

**Strengths:**

1. The paper proposes to use graph connectivity to encode transitions between memories in a Dense Associative Memory model, which includes asymmetric weights.

2. Sequential associative memories are connected to hetero-associative models.

3. Structured graphs are used to embed memories into the CDAM model (this is done for the first time to my knowledge) and a successful recall is demonstrated.

**Weaknesses:**

In the present form the paper demonstrates the possibility to write pre-defined patterns with a given graph connectivity H to CDAM and read the patterns from it. While this is of course a non-trivial capability of the model, and a first step in studying this model, it would be great to illustrate the utility of the proposed framework for some useful downstream applications that a graph community might be interested in.


**Response to Authors**

Thanks for your detailed response and revising the manuscript. I am inclined to keep my original score for this submission.

Pros:
1. Greatly improved compared to the original submission.
2. Strong results pertaining to the community detection on graphs. First, it is indeed tempting theoretically to tune parameters $a$ and $h$ in equation 1, so that the network becomes diffusive, as opposed to jumping between the memories (like in Chaudhry et al., 2023). Second, it works empirically. The results presented in figure 2 (4th panel), and figure 3 (3rd panel) are very strong.  Additionally, it is a novel (and non-trivial) idea to use associative memory for the community detection task on graphs.
3.  Experiments with video frames look very strong too.

Cons:
1. The presentation could've been cleaner. Particularly, when it comes to mathematics.
2. I do not follow the derivation of the energy. This part needs to be cleared.

**Questions:**

1. Could the authors imagine this network being helpful for some graph-related tasks that ML or graph community might be interested in?

2. I am not sure I understand how the model uses the "transition" term (with the matrix H in equation 2) in situations when one node on the graph is connected to 2 other nodes. How does the model decide which of the two connected nodes it should give more weight and transition to?

3. I am a little skeptical about using the model with negative $\beta$. In that case the softmax in equation 2 will pick the pattern with the smallest overlap with the current state $S$. Why would this setting be useful?

4. Do the authors see any interesting phenomena for $b$ in the range [0,1]? When $b$ is less than 0, as in most case studies presented, the model needs not only jump to the next pattern (on the graph), but also unlearn the current pattern. Technically, there is nothing wrong with this, but keeping $b$ in the range [0,1] would be more natural.

---

> ### Author Response · Authors · 2023-11-23
> **Reply to Reviewer K2Ug**
>
> We wish to express our thanks to Reviewer K2Ug for their productive and kind comments, which we have used to extend and revise the paper’s contents. Here we reply to their suggestions and questions.
>
> *Weaknesses*
>
> Reviewer K2Ug said, `it would be great to illustrate the utility of the proposed framework for some useful downstream applications that a graph community might be interested in.`
>
> Generally speaking, applying associative memory models directly to real-world problems is inadvisable where domain-specific models exist. It is common, however, for researchers to marry existing, e.g., gradient-based, models with associative memory models, e.g., Saha et al. ‘End-to-end Differentiable Clustering with Associative Memories’ ICML 2023 and Auer et al. ‘Conformal Prediction for Time Series with Modern Hopfield Networks’ NeurIPS 2023. Such integrations constitute independent works in their own right, and are beyond the scope of our paper, which aims to investigate associative memory networks at a more fundamental level and from a more theoretical perspective, e.g., similar to Chaudhry et al. ‘Long Sequence Hopfield Memory’ NeurIPS 2023. Nevertheless, we have added theoretical and numerical analysis to show connections between our model and graph theory concepts and techniques, including a graph clustering experiment.
>
> *Questions*
>
> 1. `Could the authors imagine this network being helpful for some graph-related tasks that ML or graph community might be interested in?`
>
> Yes, we could. As mentioned above, we believe exploring this fully is outside of this first step’s scope. However, we have included theoretical and numerical analyses to help deepen the connection to graph-related tasks, such as graph clustering experiment and temporal sequence recall.
>
> 2.  `How does the model decide which of the two connected nodes it should give more weight and transition to?`
>
> This is decided by the graph’s (weighted) adjacency matrix. In the case of an unweighted graph, all connections have an equal, arbitrary value (conventionally $1$), and so from an active node the weighted transition is equally spread to all connected nodes. However, our method can also accommodate weighted graphs (by letting the adjacency matrix elements take values from $\mathbb{R}$), in which case the transitions can be unequal. As an example, we demonstrate our method applied to a weighted graph.
>
> 3. `Why would [setting $\beta<0$] be useful?`
>
> We view this as useful only to aid theoretical understanding of the components and dynamics of dense associative memory networks and related models. When $\beta<0$, we can interpret this from the neuroscience perspective as an inversion of contributions to the excitatory-inhibitory balance, which is highly studied and important for a range of neuropsychiatric disorders, such as autism, epilepsy, and schizophrenia.
>
> 4. Reviewer K2Ug asked some questions relating to the parametrization of the model with $b$. Our initial inclination was to construct our model such that we could rigorously test it while varying this single parameter. However, as the reviewers noted, this becomes less natural outside the range of $b \in [0,1]$. So, while parsimony can offer advantages, in this case we agree it does not. We therefore separated the auto- and hetero-associative strengths into two parameters – $a$ for the auto-associative weighting and $h$ for the hetero-associative weighting. This reparametrization aids in interpretation of the model and avoids the unnaturalness present in the prior use of $b$.

---

### Official Review · Reviewer_uRxw · 2023-11-01

**Soundness:** 3 good
**Presentation:** 3 good
**Contribution:** 3 good
**Rating:** 6
**Confidence:** 3

**Summary:**

This paper proposes "Correlated Dense Associative Memories" using an iterative update defined by the linear combination of an auto-associative signal and a hetero-associative signal. We can weight this linear combination strongly in favor of the hetero-associative signal to retrieve distinct and meaningful signals.

**Strengths:**

## Experiments clearly show method effectiveness on small data

- (++) The problem in the paper is clearly introduced and defended, and the simulated experiments justify the mathematical design proposed in Eq. (2) on simulated data.
- (+) The experiments an interpretation of Dense Associative Memory that can be used for temporal sequence modeling, a problem where Transformer architectures are currently king.

**Originality**: This paper is novel to my knowledge and tackles the problem of hetero-associative memory when you are constrained to undirected graphs.

**Quality and Clarity**: This paper is well written and clear.

**Significance**: The contributions in this paper are primarily on simulated, small data, but touches an important problem in the field at large.

**Weaknesses:**

## Minimal experiments on real data; experiments primarily analyze effect of $b$

1. (- -) The paper does not include experiments on real temporal data. It is thus unclear how such a system could be applied to real problems.
2. (- -) The paper is missing analysis of the hetero-association dynamics as $t \rightarrow \infty$ at different values of $b$ (i.e., why did you choose to display results only at $t=10$?). Will the state blow up or converge to meaningful values?
3. (-) Experiments only show correlations, not the actual state itself. I feel that the hidden state $S^{(t)}$ is blowing up in value and cannot be run at low values of $b$ for too long
4. (-) Missing experiments that would consider alternative formulations of anti-hebbian modulation. Why should anti-hebbian modulation also amplify the signal from hetero-association? See Question 1.

**Questions:**

1. Eq. (2) presents the linear combination between auto- and hetero-associative memory using the hyperparameter $b$ and $(1-b)$. But this notation hints at convex combination $b \in [0, 1]$, where $b=0$ implies only hetero-associative memory and $b=1$ implies only auto-associative memory. **Why do we need a full linear combination** instead of the more intuitive convex combination? Many results require $b<<0$ (which is what the authors call "anti-hebbian" modulation) to achieve meaningful results, which both inverts the signal from auto-associative recall and amplifies the signal from hetero-associative recall. Can the authors please clarify why we need such negative values for $b$? Could the same effect be obtained by using $b=0$ and running dynamics for more steps?

2. The Zachary Karate club example hints that this is a possible method for unsupervised clustering of data. Could the authors clarify how this method could be useful for generating clusters from data, if at all?


Typos:
- Sec 3.1 paragraph 1: "As we increase b" -> "As we decrease b"

---

> ### Author Response · Authors · 2023-11-23
> **Reply to Reviewer uRxw**
>
> We thank Reviewer uRxw for their constructive and helpful comments, which has helped us significantly improve the paper. Below we respond to points they raised and their questions.
>
> *Strengths*
>
> We appreciate Reviewer uRxw’s comment on the originality of our work. In particular, they mentioned that `This paper is novel to my knowledge and tackles the problem of hetero-associative memory when you are constrained to undirected graphs.` We would like to add that our method also handles directed and weighted graphs by letting the graph’s adjacency matrix be asymmetric and letting its elements take values from $\mathbb{R}$, respectively. To emphasise this point, in the latest revision we include an example of a weighted CDAM.
>
> *Weaknesses*
>
> 1. Indeed, our original submission only used simulated data, which was a limitation. In our revision, we include new experiments with real data.
>
> 2. & 3. For practical applications, it is helpful if the network does not require many updates, which is why our initial work focussed on analysing the network states after $t=10$ updates. Nevertheless, it is still helpful to know about the long-term dynamics. We therefore added analysis of long-term dynamics both through numerical simulations and theoretical analysis.
>
> 4. Our initial inclination was to construct our model such that we could rigorously test it while varying a single parameter, $b$. However, as the reviewers noted, this became less natural outside the range of $b \in [0,1]$. So, while parsimony of this kind has its advantages, we completely agree with the reviewers that it offers little advantage in this case. We therefore separated the auto- and hetero-associative strengths into two parameters – $a$ for the auto-associative weighting and $h$ for the hetero-associative weighting. This reparameterization further aided us in the theoretical analysis of the model.
>
> *Questions*
>
> 1. Related to point 4 in weaknesses, Reviewer uRxw asked questions about the (now former) $b$ parameter. Although we have now replaced $b$ with two separate parameters ($a$ for auto- and $h$ for hetero-association), we attempt to answer these in the context of the updated parameterization.
>
> a) `Why do we need a full linear combination instead of the more intuitive convex combination?`
>
> Because otherwise the projection $Q$ becomes amplitude-limited (without increasing $\eta$ or simulating for more time-steps). This is especially true when dealing with an unweighted memory graph.
>
> b) `Can the authors please clarify why we need such negative values for $b$?`
>
> This was for the same reason as 1a above, however also because the previous analyses limited themselves to only analysing after 10 timesteps, and with small $\beta$ and $\eta$ values, making the dynamics slow. In the latest revision, we increased $\beta$ and $\eta$ to speed up dynamics where appropriate for the analyses, and in such cases such high values of $h$ (formerly negative values of $b$) are not needed.
>
> c) `Could the same effect be obtained by using $b=0$ and running dynamics for more steps?`
>
> Yes, exactly. To illustrate this more clearly, we have included results showing how choices of $a$ and $h$ relate to the temporal dynamics in the thermodynamic limit.
>
> 2. `Could the authors clarify how this method could be useful for generating clusters from data, if at all?`
>
> Thank you for the suggestion to apply our method to data clustering. We have included a new experiment to compare our method with standard graph clustering methods. We hasten to add that applying associative memory models directly to real-world problems is normally inadvisable where domain-specific models exist. It is common, however, for researchers to marry existing, e.g., gradient-based, models with associative memory models, e.g., Saha et al. ‘End-to-end Differentiable Clustering with Associative Memories’ ICML 2023 and Auer et al. ‘Conformal Prediction for Time Series with Modern Hopfield Networks’ NeurIPS 2023. Such integrations constitute independent works in their own right, and are beyond the scope of our paper, which aims to investigate associative memory networks at a more fundamental level and from a more theoretical perspective, e.g., similar to Chaudhry et al. ‘Long Sequence Hopfield Memory’ NeurIPS 2023. Nevertheless, we acknowledge that it is helpful for applied researchers to see examples of potential applications (even if those examples do not show state-of-the-art results), which is why we have added the data clustering experiment.

---

### Author Response · Authors · 2023-11-23
**General comment**

We sincerely thank all reviewers for volunteering their time to kindly help us improve this paper. We are committed to executing on the expert feedback we have received, and are happy to incorporate further suggestions or requested changes. In this general response, we summarise the changes and additions made in our revision. In separate responses, we also address specific points raised by each reviewer.

*Summary of changes and additions*

- Reparameterized the model to control auto- and hetero-association separately.
- Theoretically and numerically analysed the reparameterized model’s (long-term) dynamics.
- Deepened the connection to graph theory by analysing how the degree of vertices in the memory graph relate to the parameters and global dynamics and demonstrating (hierarchical) graph clustering.
- Conducted new experiments with real video data and replicated data from a classical neuroscience experiment.
Overhauled large portions of the text to better explain the motivations, experimental methodology, findings, and implications for both neuroscience and machine learning.

---

### Meta-Review · Area_Chair_rSH2 · 2023-12-11

**Metareview:**

This paper introduces a new associative memory model Correlated Dense Associative Memory (CDAM), which stores graph-correlated, continuous-valued, and structured memories, and exhibits hierarchical segmentation, oriented recall, and stable temporal sequence memory. The authors explore the utility of this approach in various applications, such as community detection in graphs.

The paper received mixed reviews. Reviewers commend the novel application of associative memory for community detection on graphs. However, concerns were raised about the clarity of the presentation, particularly in the mathematical formulation, and the derivation of the energy function. Some reviewers suggested the need for a clearer explanation of the practical applications of CDAM and a more thorough exploration of its implications for machine learning and neurobiology.

The authors addressed these concerns in their revisions, however despite these efforts, some reviewers remained unconvinced.

**Justification For Why Not Higher Score:**

While the paper presents a promising new approach in the field of associative memory networks, further refinement and clearer demonstration of its practical applications are necessary to fully convince the ICLR reviewers.

**Justification For Why Not Lower Score:**

N/A

---

### Decision · Program_Chairs · 2024-01-16

Reject